# Evolutionary dynamics of incubation periods

**Bertrand Ottino-Loffler[1], Jacob G Scott[2,3], Steven H Strogatz[1]\***

[1]Center for Applied Mathematics, Cornell University, Ithaca, United States; [2]Department of Translational Hematology and Oncology Research , Cleveland Clinic, Cleveland, United States; [3]Department of Radiation Oncology, Cleveland Clinic, Cleveland, United States

**Abstract** The incubation period for typhoid, polio, measles, leukemia and many other diseases follows a right-skewed, approximately lognormal distribution. Although this pattern was discovered more than sixty years ago, it remains an open question to explain its ubiquity. Here, we propose an explanation based on evolutionary dynamics on graphs. For simple models of a mutant or pathogen invading a network-structured population of healthy cells, we show that skewed distributions of incubation periods emerge for a wide range of assumptions about invader fitness, competition dynamics, and network structure. The skewness stems from stochastic mechanisms associated with two classic problems in probability theory: the coupon collector and the random walk. Unlike previous explanations that rely crucially on heterogeneity, our results hold even for homogeneous populations. Thus, we predict that two equally healthy individuals subjected to equal doses of equally pathogenic agents may, by chance alone, show remarkably different time courses of disease.

DOI: https://doi.org/10.7554/eLife.30212.001

## Introduction

The discovery that incubation periods tend to follow right-skewed distributions originally came from epidemiological investigations of incidents in which many people were simultaneously and inadvertently exposed to a pathogen. For example, at a church dinner in Hanford, California on March 17, 1914, ninety-three individuals became infected with typhoid fever after eating contaminated spaghetti prepared by an asymptomatic carrier known to posterity as Mrs. X. Using the known time of exposure and onset of symptoms for the 93 cases, *Sawyer, 1914* found that the incubation periods ranged from 3 to 29 days, with a mode of only 6 days and a distribution that was strongly skewed to the right. Similar results were later found for other infectious diseases. Surveying the literature in 1950, Sartwell noted a striking pattern: the incubation periods of diseases as diverse as streptococcal sore throat (*Sartwell, 1950*) (*Figure 1a*), measles (*Stillerman and Thalhimer, 1944*), polio, malaria, chicken pox, and the common cold were all, to a good approximation, lognormally distributed (*Sartwell, 1950*). On a time scale of years instead of days, the incubation periods for bladder cancer (*Goldblatt, 1949*) (*Figure 1b*), skin cancer, radiation-induced leukemia, and other cancers were also found to be approximately lognormally distributed (*Armenian and Lilienfeld, 1974*).

Two natural questions arise: Why should incubation periods be distributed at all, and why should they be distributed in the same way for different diseases? Previous explanations rest on the presumed heterogeneity of the host, the pathogen, or the dose (*Sartwell, 1950*; *Nishiura, 2007*; *Horner and Samsa, 1992*). To see how this works, return to the typhoid outbreak at the Hanford church dinner (*Sawyer, 1914*). Every person who ate that spaghetti presumably had a different level of overall health and immune function, and every plate of spaghetti was likely contaminated with a different dose and possibly even strain of typhoid.

**\*For correspondence:** strogatz@cornell.edu

**Competing interests:** The authors declare that no competing interests exist.

**eLife digest** When one child goes to school with a throat infection, many of his or her classmates will often start to come down with a sore throat after two or three days. A few of the children will get sick sooner, the very next day, while others may take about a week. As such, there is a distribution of incubation periods – the time from exposure to illness – across the children in the class.

When plotted on a graph, the distribution of incubation periods is not the normal bell curve. Rather the curve looks lopsided, with a long tail on the right. Plotting the logarithms of the incubation periods, however, rather than the incubation periods themselves, does give a normal distribution. As such, statisticians refer to this kind of curve as a "lognormal distribution". Remarkably, many other, completely unrelated, diseases – like typhoid fever or bladder cancer – also have approximately lognormal distributions of incubation periods. This raised the question: why do such different diseases show such a similar curve?

Working with a simple mathematical model in which chance plays a key role, Ottino-Löffler et al. calculate how long it takes for a bacterial infection or cancer cell to take over a network of healthy cells. The model explains why a lognormal-like distribution of incubation periods, modeled as takeover times, is so ubiquitous. It emerges from the random dynamics of the incubation process itself, as the disease-causing microbe or mutant cancer cell competes with the cells of the host.

Intuitively, this new analysis builds on insights from the "coupon collector's problem": a classical problem in mathematics that describes the situation where a person collects items like baseball cards, stamps, or cartoon monsters in a videogame. If a random item arrives every day, and the collector's luck is bad, they may have to wait a long time to collect those last few items. Similarly, in the model of Ottino-Löffler et al., the takeover time is dominated by dramatic slowdowns near the start or end of the infection process. These effects lead to an approximately lognormal distribution, with long waits, as seen in so many diseases.

Ottino-Löffler et al. do not anticipate that their findings will have direct benefits for medicine or public health. Instead, they believe their results could help to advance basic research in the fields of epidemiology, evolutionary biology and cancer research. The findings might also make an impact outside biology. The term "contagion" has now become a familiar metaphor for the spread of everything from computer viruses to bank failures. This model sheds light on how long it takes for a contagion to take over a network, for a variety of idealized networks and spreading processes.

DOI: https://doi.org/10.7554/eLife.30212.002

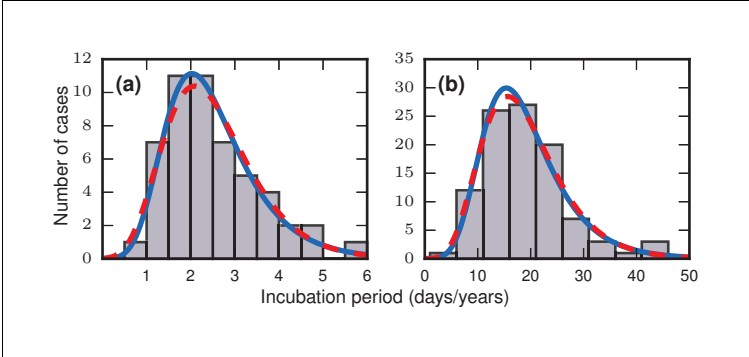

**Figure 1.** Frequency distributions of incubation periods for two diseases. Data redrawn from historic examples. Dashed red curves are noncentral lognormal distributions. Solid blue curves are Gumbel distributions, predicted by the theory developed here. Both sets of curves were fitted via the method of moments. (a) Data from an outbreak of food-borne streptococcal sore throat, reported in 1950 (*Sartwell, 1950*). Time is measured in units of days. (b) Data from a 1949 study of bladder tumors among workers following occupational exposure to a carcinogen in a dye plant (*Goldblatt, 1949*). Time is measured in units of years.

DOI: https://doi.org/10.7554/eLife.30212.003

## Box 1. Dispersion factors.

The **Dispersion Factor** of a distribution or dataset is defined to be its geometric standard deviation. Or more explicitly, given a positive dataset $x_n$, it is $\sigma_G$, where

$$\mu_G := \exp\left[\sum_n \frac{1}{N}\log(x_n)\right]$$

$$\sigma_G := \exp\left[\sum_n \frac{1}{N-1}\log(x_n/\mu_G)^2\right].$$

We measure this quantity for multiple reasons. While $\mu_G$ is a dimensionful quantity, $\sigma_G$ is dimensionless. Secondly, $\log(\sigma_G)$ is the maximum likelihood estimator for the scale parameter of an unshifted lognormal distribution. Moreover, this is the quantity Sartwell used to describe the variability of incubation periods (**Sartwell, 1950**), so it is a useful point of comparison.
DOI: https://doi.org/10.7554/eLife.30212.004

Suppose the typhoid bacteria proliferated exponentially fast within the hosts and triggered symptoms when they reached a fixed threshold. Then, if the bacterial dose, growth rate, or triggering threshold were normally distributed across the hosts, one can show that the resulting distribution of incubation periods would have been either exactly or approximately lognormal (see Results, 'Influence of heterogeneity'). On the other hand, there is counter-evidence that lognormal distributions can occur even if some of these sources of heterogeneity are lacking. For example, *Sartwell, 1950* reanalyzed data from a study (*Bodian et al., 1949*) in which identical doses and strains of polio virus were injected into the brains of hundreds of rhesus monkeys. The incubation period, defined as the time from inoculation to the onset of paralysis, was still found to be approximately lognormally distributed, even though the route of infection and the viral dose and strain were held constant. Moreover, the lognormal distributions commonly observed for human diseases have a particular shape, with a dispersion factor (*Sartwell, 1950*) around $1.1 - 1.5$, which previous models cannot explain without special parameter tuning. (See *Box 1* for the definition of dispersion factors.)

Here, we propose a new explanation for the skewed distribution of incubation periods. Instead of heterogeneity, it relies on the stochastic dynamics of the incubation process, as the pathogen invades, multiplies, and competes with itself and the cells of the host in a structured network topology. The theory predicts that under a broad range of circumstances, incubation periods should follow a right-skewed distribution that resembles a lognormal, but is actually a Gumbel, one of the universal extreme value distributions (*Kotz and Nadarajah, 2000*). Heterogeneity is not required, but it is allowed; it does not qualitatively alter our results when included.

## Results

### Mathematical Model

We model the incubation process using the formalism of evolutionary graph theory (*Lieberman et al., 2005*; *Nowak, 2006*; *Ohtsuki et al., 2006*; *Ashcroft et al., 2015*). A network of $N \gg 1$ nodes is used to represent an environment within a host where a pathogenic agent, such as a harmful bacterium or a cancer cell, is invading and reproducing. The network could represent several plausible biological scenarios, for example the intestinal microbiome, where harmful typhoid bacteria are competing against a benign resident population of gut flora in a mixing system (modeled as a complete graph); or it could represent mutated leukemic stem-cells vying for space against healthy hematopoietic stem cells within the well-organized three-dimensional bone marrow space (modeled as a 3D lattice); or a flat epithelial sheet with an early squamous cancer compromising and invading nearby healthy cells (modeled as a 2D lattice). For the sake of generality, we will refer to the two types of agents as healthy residents and harmful invaders.

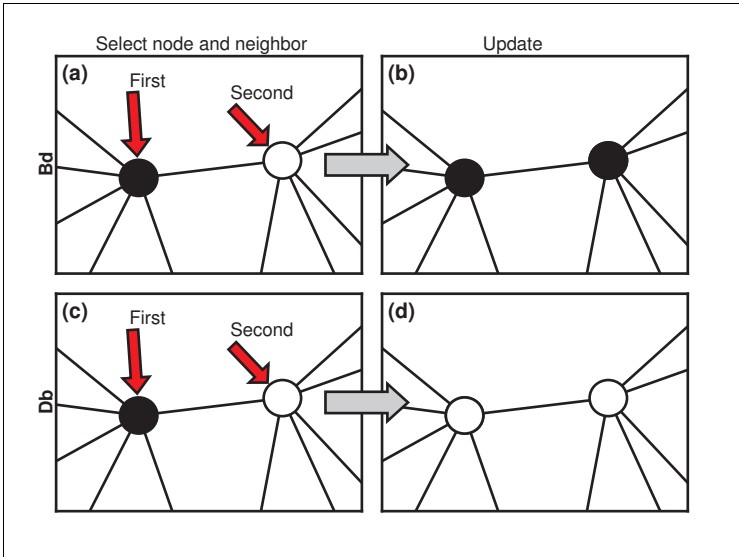

**Figure 2.** Evolutionary update rules. (**a**) In the Birth-death (Bd) update rule, a node anywhere in the network is selected at random, with probability proportional to its fitness, and one of its neighbors is selected at random, uniformly. (**b**) The neighbor takes on the type of the first node. In biological terms, one can interpret this rule in two ways: either the first node transforms the second; or it gives birth to an identical offspring that replaces the second. (**c**) In the Death-birth (Db) update rule, a node is selected at random to die, with probability inversely proportional to its fitness, and one of its neighbors is selected at random, uniformly, to give birth to one offspring. (**d**) The first node is replaced by the offspring of the second.

DOI: https://doi.org/10.7554/eLife.30212.005

## Box 2. Nomenclature for the Moran model.

There are many distinct variations on the basic Moran model, but the six most popular all consist of a two-step update. The first step selects a node over the entire population, whereas the second step selects a neighboring node from the neighbors of the first.

However, the content of each step can vary from one model to another. For example, the selection in each step can occur in a fitness-weighted fashion. Also, the node in the first step can be interpreted as either giving birth, or dying.

To avoid confusion, we use standard abbreviations to distinguish the different models, as illustrated by the table below. The order of letters indicates the order of the steps, and the capitalization denotes which steps have a fitness dependence. For example, dB refers to the update rule where the first step uniformly selects a node from the entire population to die, and then one of its neighbors is selected, with probability proportional to fitness, to replace it. In this paper, we focus exclusively on Bd and Db. Although much the same analysis could be done on any of the other update rules, these two were showcased due to their relative popularity and simplicity.

| Step order | Fitness-step first | Fitness-step second | Both fitness-steps |
|---|---|---|---|
| **Birth first** | Bd | bD | BD |
| **Death first** | Db | dB | DB |

DOI: https://doi.org/10.7554/eLife.30212.006

While Sartwell's law has been applied to many different types of diseases with diverse etiologies, the model we propose makes the most sense for asexually reproducing invaders, like cancer cells or bacteria. Viruses, on the other hand, often reproduce with a 'one-to-many' dynamic, which is not faithfully captured in this model. So, while the general phenomenon of network invasion seems to apply to viruses as well, the model in its present form is not well suited to describe their dynamics.

Considering asexually reproducing and competing invaders, then, we choose to model the invasion dynamics as a Moran process (*Moran, 1958*; *Williams and Bjerknes, 1972*; *Lieberman et al., 2005*; *Nowak, 2006*). Invaders are assigned a relative fitness $r$ (suggestively called the carcinogenic advantage by *Williams and Bjerknes, 1972*). The fitness of residents is normalized to 1. We consider two versions of the Moran process. In the Birth-death (Bd) version (*Figure 2a*), a random node is chosen, with probability proportional to its fitness. It gives birth to a single offspring. Then, one of its neighbors is chosen uniformly at random to die and is replaced by the offspring (*Figure 2b*). We also consider Death-birth (Db) updates (*Figure 2c,d*). In this version of the model, a node is randomly selected for death, with probability proportional to $1/r$; then a copy of a uniformly random neighbor replaces it. To test the robustness of our results, we study both versions of the Moran model on various networks: complete graphs, star graphs, Erdős-Rényi random graphs, one-, two-, and three-dimensional lattices, and small-world, scale-free, and $k$-regular networks. We also vary the invader fitness $r$ and the model criterion for the onset of symptoms. These extensions are presented in the Materials and methods, Figures 5, 6. *Box 2* discusses other variants of the Moran model. Here we focus on the simplest cases to elucidate the basic mechanisms.

Our simulations start with a single invader placed at a random node in a network of otherwise healthy residents. The update rule is applied at discrete time steps. In the long run, either the invaders replace all the residents, or vice versa. If symptoms are triggered when the entire network has been taken over by invaders, then the incubation period is the number of time steps between the introduction of the invader and its fixation. On the other hand, if the invaders die out and the healthy cells take over, then the process is stopped and no observable symptoms manifest. Later, in the paper, we consider a generalization from complete to partial takeovers, but for now the incubation period will refer to a complete takeover.

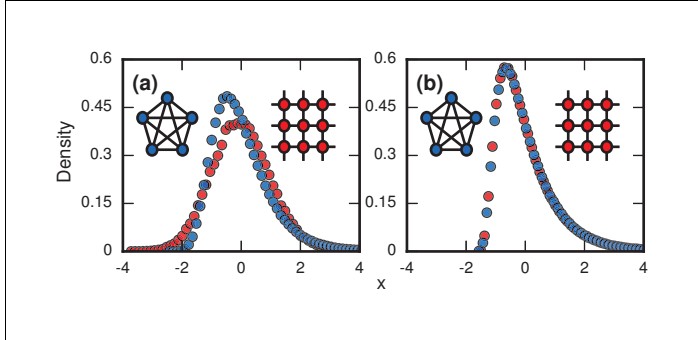

**Figure 3.** Network topology and invader fitness shape the distribution of incubation periods. Plots show simulated distributions of incubation periods, defined here as invader fixation times. Starting from a single invader at a random node, the state of the network was updated by Birth-death dynamics on both a complete graph and a two-dimensional (2D) lattice. Results for the Death-birth update rule (not shown) are identical. All distributions are normalized to have zero mean and unit variance. (a) Infinitely fit invader. For invader fitness $r \to \infty$, the distribution is right-skewed for a complete graph (blue symbols). It approaches a Gumbel distribution as $N \to \infty$, where $N$ is the number of nodes in the network. In contrast, for a 2D lattice (red symbols) the incubation periods are normally distributed. The difference is that a coupon collection mechanism operates in the complete graph and in lattices of sufficiently high dimension $d \geq 3$; this mechanism causes the right skew. Simulations used $10^6$ repetitions on a complete graph of $N = 150$ nodes, and $10^5$ repetitions for a 2D lattice of $N = 30^2$ nodes. (b) Neutrally fit invader. Distributions of incubation periods are shown for invader fitness $r = 1$, using $10^6$ repetitions on a complete graph of $N = 50$ nodes (blue symbols), and $10^5$ repetitions for a 2D lattice of $N = 7^2$ nodes (red symbols). Similar right-skewed distributions occur for both networks, caused by a conditioned random walk mechanism.
DOI: https://doi.org/10.7554/eLife.30212.007

Our notion of time in this model is linked directly to the biology of invasion of a reproducing asexual pathogen that divides and replaces other cells sequentially. Instead of considering divisions as a rate, and therefore linking the dynamics to real time, we consider time steps to be individual division events. This is more akin to the standard methods of modeling chemical interactions, as in the Gillespie algorithm (*Gillespie, 1977*). This focus on the biology of the individual pathogen (or cancer cell) also provides a simple explanation for how diseases with very different natural histories can have the same analytic distribution of incubation time. As each different disease would have a different characteristic mean doubling time, while the shape of the distributions might be the same, the physical time taken would scale with the characteristic proliferation time. Future iterations of this model could consider deriving an exact scaling between physical time and this biological event-based updating of time.

## Infinitely fit invaders

First, consider what happens if the invaders have infinite fitness ($r \to \infty$) in the Birth-death model. While an exaggeration, this case is instructive and is a reasonable approximation for aggressive cancers and infections. In this limit, the dynamics simplify enormously: only the invaders reproduce. But because they give birth and replace their neighbors blindly, they waste time whenever they compete between themselves and one invader replaces another. These random self-replacements slow down the incubation process, and make it highly variable. In fact, the level of in-fighting is what determines the incubation period in this case. Beyond fitness, the topology of the network matters too. For low-dimensional networks, exemplified by a two-dimensional lattice (*Figure 3a* , red circles), the growth rate of the invader population remains roughly constant as takeover occurs. This leads to a normal distribution of incubation periods (*Figure 3a*, red circles; and see Methods and Materials, 'Birth-death, other solvable networks'). However, on very high-dimensional networks like the complete graph (*Figure 3a*, blue circles), the distribution becomes right skewed. Intuitively, this happens because every invader now has a chance of replacing any healthy node or any other invader. It is as if at every time step a candidate node for replacement gets blindly drawn from a bag, relabeled as an invader, and returned to the bag. At the start of the incubation process, almost every draw adds another invader to the population and the infection progresses rapidly. But near the end, it will take many, many draws to blindly fish out the last remaining healthy node, as needed to terminate the incubation period. This slowing-down phenomenon near the end should feel familiar to anyone who has tried to complete a collection of baseball cards, stamps, or coupons, since they are all manifestations of the coupon collector's problem, a well-studied concept in probability theory (*Pósfai, 2010*; *Feller, 1968*; *Erdős and Rényi, 1961*). Because of those frustratingly long waits to collect the final healthy node, the incubation period distribution gets skewed to the right. In the infinite-$N$ limit (see Methods and Materials, 'Birth-death, complete graph'), the coupon collector's process returns a Gumbel distribution, which resembles a lognormal and can be mistaken for it (*Read, 1998*). Indeed, when a Gumbel and a lognormal are fit to the same real data, as in *Figure 1*, it is hard to tell them apart. All this analysis can easily be repeated for the Death-birth model with minimal changes.

## Neutrally fit invaders

At the other extreme, suppose the invaders have no selective advantage ($r = 1$). Then a different stochastic mechanism skews the distribution of incubation periods to the right (*Figure 3b* and Methods and Materials, 'Random Walk Skewness'). For many networks, the dynamics reduce to an unbiased random walk on the number of invaders, with waiting times at each population level. There are two absorbing states, corresponding to both 00 and $N$ invaders for the two kinds of fixation. However, we only care about random walks that successfully hit $N$, as these represent disease processes that manifest symptoms, so we must always condition on its success. This demands that the invader experience early success and growth, pushing it away from probable extinction. This conditioning introduces a bias that makes short incubation times probable, but long walks may still occasionally occur, driving the mean time above the median. In short, a conditioned random walk will introduce a right skew in the distribution of incubation periods. This effect holds for both high- and low-dimensional networks (*Figure 3b*), and for Birth-death and Death-birth dynamics.

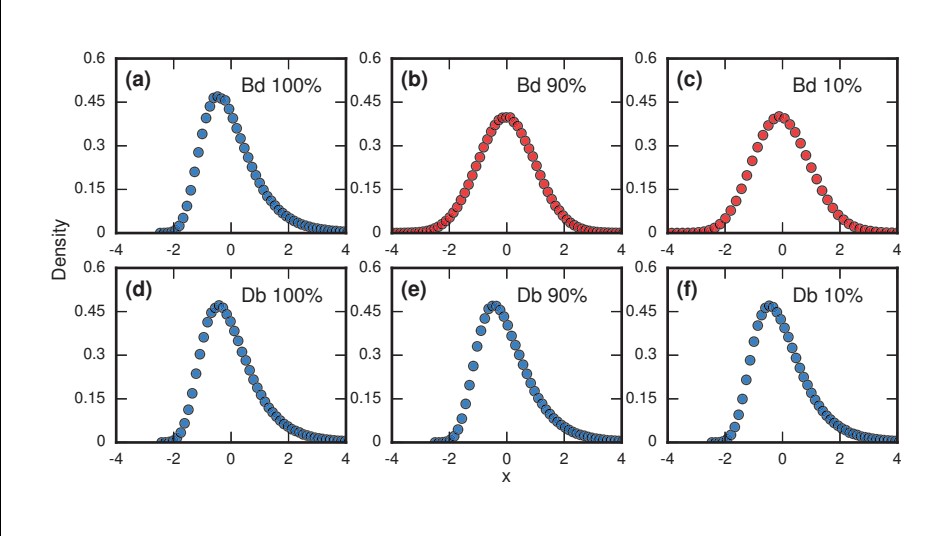

**Figure 4.** Testing robustness. The plots show how the distribution of incubation periods does or does not change when we modify the assumed update rules and criterion for the onset of symptoms. Both Birth-death (Bd) and Death-birth (Db) dynamics were simulated on a complete graph of $N = 5000$ nodes, using a infinite invader fitness. Incubation periods are now defined as times needed for invaders to take over a fraction $f$ of the whole network. All distributions are normalized to have zero mean and unit variance. Data points are color-coded according to the nature of the distribution: blue indicates a Gumbel distribution, and red indicates a normal distribution. (a) The distribution of times till invader fixation ($f = 1$) under Birth-death dynamics. The Gumbel distribution of *Figure 3a* persists. (b) When $f$ is reduced to 0.9, the incubation periods under Birth-death dynamics become normally distributed instead of skewed. (c) When $f$ is reduced to 0.1, the incubation period distribution remains normally distributed. By contrast, Death-birth dynamics are insensitive to this modification: the Gumbel distribution persists not only for (d) $f = 1$ but even for (e) $f = 0.9$ and (f) $f = 0.1$. The difference in sensitivity between the two types of dynamics can be explained intuitively by when the slowest part of the coupon collection process occurs. For Death-birth dynamics, it occurs near the *beginning* of the invasion, when it takes a long time to randomly select one of the few available invaders to give birth. Since the slow part of coupon collection occurs near the beginning, it is insensitive to the end-condition $f<1$. In contrast, the slow part occurs near the *end* of the invasion for Birth-death dynamics (when residents are scarce), and hence gets truncated when $f<1$, giving rise to a normal instead of a right-skewed distribution.

DOI: https://doi.org/10.7554/eLife.30212.008

## Testing robustness to update rule and truncation

Right-skewed distributions typically persist in the face of various perturbations to the model, but some perturbations can turn them into normal distributions. For example, suppose we allow symptoms to occur when invaders take over only a fraction $f$ of the whole network. This is a reasonable consideration as leukemic cells need not take over all the bone marrow before leukemia becomes evident, nor does typhoid need to overwhelm all the cells in the microbiome before causing fever; indeed it is likely far fewer in both cases. *Figure 4* contrasts what happens for Birth-death and Death-birth dynamics under these assumptions. When $r = \infty$, the Gumbel distribution of *Figure 3a* persists for $f = 1$ (*Figure 4a*), but turns into a normal distribution (*Baum and Billingsley, 1965*) when $f = 0.9$ (*Figure 4b*) or $f = 0.1$ (*Figure 4c*). Yet under Death-birth dynamics, the distribution stays Gumbel for all nonzero values of $f$ (*Figure 4d,e,f*). The fact that birth-death dynamics returns a normal for $0<f<1$ whereas Death-birth still returns a Gumbel can be rationalized via various convergence theorems (*Baum and Billingsley, 1965*; *Ottino-Löffler et al., 2017*; *Pósfai, 2010*). However, the fact that similar update rules behave so differently under a reasonable perturbation should caution us to be mindful of our choice of models.

## Influence of heterogeneity

Historically, the distribution of incubation periods has been ascribed to heterogeneity (*Sartwell, 1950*; *Nishiura, 2007*; *Horner and Samsa, 1992*) in the fitness (growth rate, say) or dose of

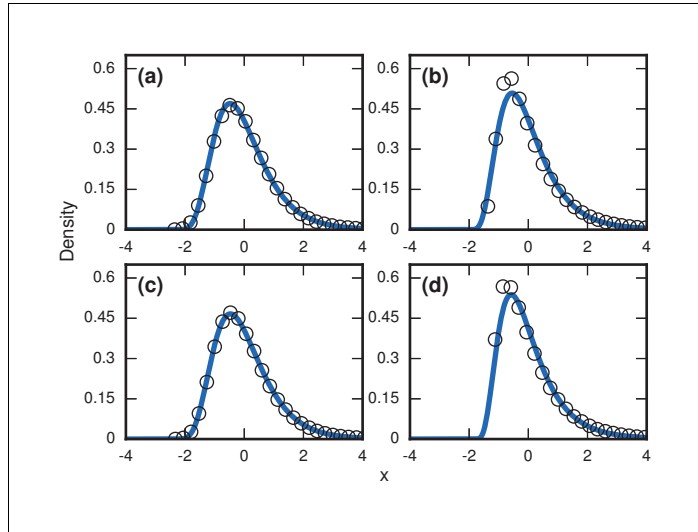

**Figure 5.** Robustness to heterogeneity. Simulated, fitted, and normalized distributions of incubation periods for birth-death dynamics on a complete graph of $N = 500$ nodes. Unless stated otherwise, each simulation used an invader fitness of $r = 10$, measured times till complete takeover ($f = 1$), and started from an initial dose of 1 invader. Runs where the dosage was not smaller than the truncation point were rejected. The blue curves indicate noncentral lognormals fitted via the method of moments. (a) Heterogeneous fitness of invader. Every run used a different $r$ selected from a Gamma distribution with a shape parameter of 10. (b) Heterogeneity of host response. Instead of waiting until all $N$ residents had been replaced by invaders, every run used a different truncation point uniformly selected from $\{2, 3, \ldots N\}$. (c) Heterogeneity of dosage. Every run had a different starting population drawn from a Poisson of mean 10 and a shift of 1. (d) Heterogeneity of invader fitness, host response, and dosage. Every run used an $r$ drawn from Gamma(10), a truncation point $f$ drawn from Uniform(0,1), and a dosage drawn from Poisson(10)+1.

DOI: https://doi.org/10.7554/eLife.30212.009

the pathogen, or in host factors like immune response. To see how these potential sources of heterogeneity could account for the skewed and approximately lognormal distribution of incubation periods, consider a pathogen growing exponentially with rate $r$ from an initial population $N_0$, so that its population at time $t$ is given by $N(t) = N_0 e^{rt}$. If an immune response or other detectable symptoms are triggered when $N$ reaches a threshold population $\theta$, then the incubation time $T$ satisfies $N(T) = N_0 e^{rT} = \theta$. Solving for $T$ yields

$$T = \frac{1}{r}(\log \theta - \log N_0). \tag{1}$$

So if either the threshold $\theta$ or the inoculum $N_0$ are normally distributed across the host population, the incubation period $T$ will be lognormally distributed. Likewise, but in a more qualitative sense, a normal distribution of pathogen growth rates $r$ will also produce a skewed distribution that resembles a lognormal (*Nishiura, 2007*). However, if there is no randomness in any of those sources, this model predicts a single deterministic value of $T$ for the incubation period.

In contrast, the stochastic model proposed here does not need these sources of heterogeneity to produce right-skewed distributions. But if they happen to be present, as they likely are for many real diseases, our model can accommodate them. Indeed, when any of the three sources of heterogeneity are included in our model, they only serve to make the predicted distributions even more right-skewed, as we now show.

First, to emulate the heterogeneity of the strength of the pathogen, we assume heterogeneity in the parameter $r$ (which, in our model, governs the fitness of the invading cells relative to those of the host). In particular we randomly draw a different $r > 0$ in each simulation, to simulate different hosts being infected with different pathogenic strains. The resulting distribution of invader fixation times depends on the distribution of the $r$'s, but our investigations demonstrate they consistently produce right-skewed distributions (*Figure 5a*).

Second, to emulate the heterogeneity of host factors like immune response, we allow variability in the parameter $f$, which quantifies the fraction of the network that needs to be invaded before symptoms appear. Let $T_f$ denote the time it takes for $N \cdot f$ of the original resident nodes to be replaced by invaders. If we draw $f$ randomly from some distribution, then essentially each host has a different threshold at which symptoms appear. In contrast to *Figure 4b*, where we saw that repeated simulations for a host population with a single, fixed, deterministic $f$ can cause skewed distributions to turn into normal distributions, that is no longer the case when heterogeneity is included, as *Figure 5b* indicates. In fact, the heterogeneity actually causes even more right-skew than before.

Third, emulating variable doses is also straightforward. Instead of always starting with a single invader cell, we choose the initial number of invaders according to some distribution. Again, this modification does not remove the right-skewed behavior established in the Moran model (*Figure 5c*).

Finally, we can apply all these sources of heterogeneity at once, and remain with a right-skewed distribution (*Figure 5d*). In summary, although our main results were obtained by analyzing stochastic models of homogeneous host and pathogen populations, allowing for heterogeneity makes the predicted right-skewed distributions more, not less, prominent.

## Discussion

The evolutionary dynamical model presented here is intended to mimic the within-host development of certain cancers and bacterial infections. It is not well suited to the dynamics of viruses. Thus, explaining why Sartwell's law also holds for so many viral diseases remains an open question.

Our model suggests two basic mechanisms underlie the observed right-skewed, approximately lognormal distributions of incubation periods. When the fitness of the pathogen is high, the skew comes from coupon collection; when the pathogen fitness is neutral or low, the skew comes from conditioned random walks; and at intermediate fitnesses, a combination of the two creates skew. Neither of these effects demand any heterogeneity from the invader or the host. However, the model can accommodate such heterogeneity, either by having the invader fitness $r$ be randomly drawn, or by having symptoms occur when a random fraction $f$ of the host network has been invaded. Our simulations show that both sources of heterogeneity only exaggerate the level of right-skewness we would have seen without them (See Results,'Influence of heterogeneity', *Figure 5*).

Beyond accounting qualitatively for the distributions of incubation periods, our model accounts for a quantitative feature that has never been explained before. As shown in Methods and Materials, *Table 1*, the distributions generated by highly fit pathogens and mutants are predicted to have

**Table 1.** Model dispersion factors.

Dispersion factors (geometric standard deviations, see *Box 1*) for the simulated distributions of incubation periods shown in *Figures 6,7,8* , for different networks and invader fitness levels $r$. Errors represent 95% confidence intervals. Due to finite size effects, the dispersion factors exceed 1 for 1D and 2D lattices with $r = \infty$ (they should approach one as $N \to \infty$). Dispersion factors for the $r = 1$ case are larger than for the $r = \infty$ case, but are more uniform for different network topologies.

| Network | $r = \infty$ | $r = 1$ |
| --- | --- | --- |
| Complete | $1.2386 \pm 0.0004$ | $1.6629 \pm 0.0012$ |
| Star | $1.3463 \pm 0.0006$ | $1.6875 \pm 0.0012$ |
| 1D Lattice | $1.1418 \pm 0.0002$ | $1.7907 \pm 0.0014$ |
| 2D Lattice | $1.0731 \pm 0.0003$ | $1.6799 \pm 0.0012$ |
| 3D Lattice | $1.1289 \pm 0.0006$ | $1.6659 \pm 0.0012$ |
| Erdős-Rényi | $1.2586 \pm 0.0004$ | $1.6900 \pm 0.0012$ |
| Small-World | $1.2604 \pm 0.0004$ | $1.7693 \pm 0.0014$ |
| k-Regular | $1.2125 \pm 0.0003$ | $1.7229 \pm 0.0013$ |
| Scale-Free | $1.4189 \pm 0.0007$ | $1.7399 \pm 0.0013$ |

DOI: https://doi.org/10.7554/eLife.30212.010

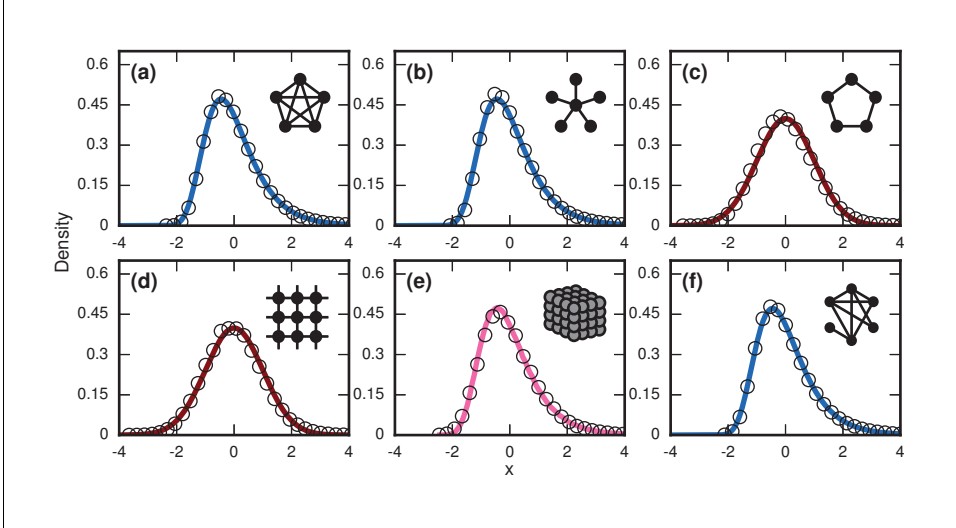

**Figure 6.** Network dependence of incubation periods. Distributions of invader fixation times, normalized to have zero mean and unit variance, are shown for infinite-$r$ Birth-death dynamics on various networks. Open circles show simulation results. Curves show analytical predictions: blue curves are Gumbels, red are normals, and pink is an intermediate distribution. Insets show schematics of networks. (a) The distribution of fixation times for a complete graph on $N = 150$ nodes, for $10^6$ runs. Distribution normalized according to analytically calculated mean and standard deviation. Curve shows a Gumbel distribution. (b) The Gumbel distribution of fixation times for a star graph with $N = 75$ spokes, for $10^6$ runs. Distribution normalized according to analytically calculated mean and standard deviation. (c) Normal distribution of fixation times for a 1D ring on $N = 75$ nodes, for $10^6$ runs. Distribution normalized according to analytically calculated mean and standard deviation. (d) Normal distribution of fixation times for a 2D lattice of $N = 60 \times 60$ nodes, for $10^5$ runs. Distribution normalized according to empirically calculated mean and standard deviation. (e) The distribution of fixation times for a 3D lattice of $N = 11^3$ nodes, for $10^5$ runs. Distribution normalized according to empirically calculated mean and standard deviation. The predicted distribution is the result of an approximating sum of exponential random variables under $10^6$ repetitions. (f) The distribution of fixation times for an Erdős-Rényi random graph on $N = 115$ nodes with an edge probability of $\rho = 0.5$. Distribution normalized according to empirically calculated mean and standard deviation.
DOI: https://doi.org/10.7554/eLife.30212.011

dispersion factors (also known as geometric standard deviations; see *Box 1*) of about $1.1 - 1.4$, close to the actual values of $1.1 - 1.5$ observed for various infectious diseases (*Sartwell, 1950*;*Sartwell, 1966*;*Nishiura, 2007*). Moreover, the model also helps to explain why so few infectious diseases yield dispersion factors greater than 1.5. Such high dispersion factors arise only for $r \approx 1$, corresponding to pathogens or mutants that are only slightly more fit than the resident populations against which they are competing.

On the other hand, it is tempting to speculate that this regime of nearly neutral fitness may be more relevant to cancer development. While it is likely that tumor cells late in the disease process have much higher fitness than healthy cells secondary to continued selection (*Scott and Marusyk, 2017*), there is ample evidence that most cancers have long latency periods, for example in genetic data from pancreatic cancers (*Yachida et al., 2010*). One could speculate that during this early period, which accounts for the majority of the cancer's time in the patient, the fitness is nearly neutral. For the cancer data reviewed by (*Armenian and Lilienfeld, 1974*), the observed distributions typically had dispersion factors around $1.4 - 1.9$. In our model, these high dispersion factors tend to arise when the invader is only slightly more fit than residents. This is also consistent with the suggestion of (*Williams and Bjerknes, 1972*); the shape of tumors in the model most closely resembled that of real tumors when the fitness of the invaders was only slightly above neutral.

In 1546, *Fracastorii, 1930* described the incubation of rabies after a bite from an rabid dog as 'stealthy, slow, and gradual.' Today, nearly five centuries later, the dynamics of incubation processes remain stealthy and slow to yield their secrets. We have tried to shed light on their patterns of variability with the help of a new conceptual tool, evolutionary graph theory. This approach provides a

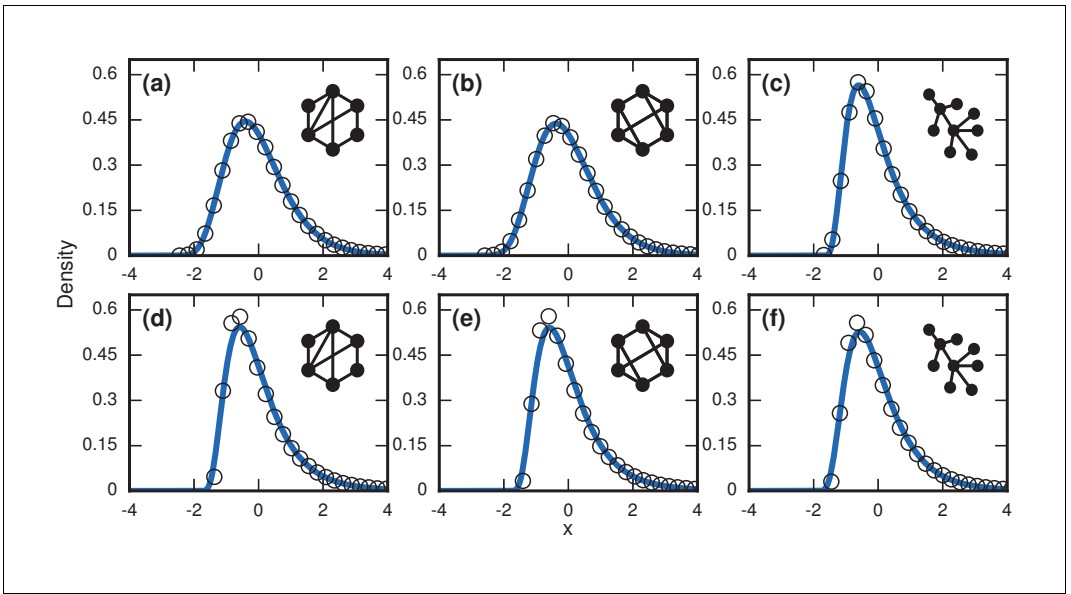

**Figure 7.** Complex networks. Simulated and fitted distributions of invader fixation times for Birth-death dynamics on small-world, scale-free, and $k$-regular networks. All distributions were normalized to have mean zero and unit variance. The curves indicate non-central lognormals fitted to the first three moments of the data. All distributions are the result of $10^6$ simulations. The figures in the top row ((**a**), (**b**), (**c**)) used invader fitness $r = \infty$, whereas the figures in the bottom row ((**d**), (**e**), (**f**)) used neutral fitness $r = 1$. (**a**) Newman-Watts-Strogatz small-world ring network with shortcut probability of $\rho = 0.25$ on $N = 75$. (**b**) Random 3-regular graph on $N = 100$ nodes. (**c**) Barabasi-Albert scale-free network with a minimum degree of 3 and $N = 100$ nodes. (**d**) Newman-Watts-Strogatz small-world ring network with shortcut probability of $\rho = 0.25$ on $N = 25$ nodes. (**e**) Random 3-regular graph on $N = 22$ nodes. (**f**) Barabasi-Albert scale-free network with a minimum degree of 3 and $N = 22$ nodes.
DOI: https://doi.org/10.7554/eLife.30212.012

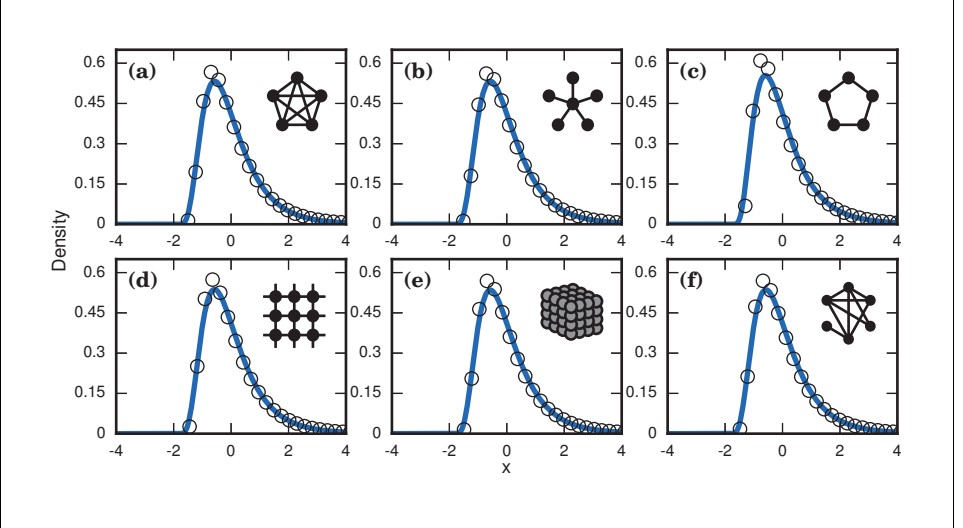

**Figure 8.** Neutrally fit invader ($r = 1$). Simulated and fitted distributions of invader fixation times are shown for Birth-death dynamics on various networks. All distributions were normalized to have mean zero and unit variance. The curves indicate noncentral lognormals fitted via the method of moments. (**a**) Complete graph on $N = 50$ nodes, for $10^6$ runs. (**b**) Star graph with $N = 25$ spokes, for $10^6$ runs. (**c**) One-dimensional ring on $N = 50$ nodes, for $10^6$ runs. (**d**) Two-dimensional lattice on $N = 7 \times 7$ nodes, for $10^6$ runs. (**e**) Three-dimensional lattice on $N = 4^3$ nodes for $10^6$ runs. (**f**) Erdős-Rényi random graph on $N = 25$ nodes with an edge probability of $\rho = 0.5$.
DOI: https://doi.org/10.7554/eLife.30212.013

possible solution to the longstanding question of why so many disparate diseases show such similarly-shaped distributions of incubation periods. What remains is to quantify the dynamics of incubation processes experimentally with high-resolution measurements in time and space.

Aside from their possible application to incubation processes, our results also shed light on a broader theoretical question in evolutionary dynamics: when a mutant invades a structured population of residents, how does the distribution of mutant fixation times depend on the network structure of the population? Early work in evolutionary graph theory (*Lieberman et al., 2005*; *Nowak, 2006*; *Ohtsuki et al., 2006*) concentrated on the network's impact on the probability of mutant fixation and the mean time to fixation. More recent studies have gone beyond the mean time to consider the full distribution of fixation times (*Ashcroft et al., 2015*), as we have also done here. We hope that our exact results for disparate topologies and dynamics will stimulate further investigations of these important questions in evolutionary biology.

## Materials and methods

Here we describe the model and our analytical and numerical results in further detail. We also test the robustness of our claims with respect to relaxation of the various assumptions in the model. See the Appendix for complete proofs of analytical results.

### Birth-death, complete graph

The population of cells is represented by a network of $N$ nodes. Edges between nodes indicate which cells can potentially interact with each other. There are two types of cells: harmful invaders with fitness $r$, and healthy residents with fitness 1. All simulations are initialized with a single invader placed at a random node.

The Moran Birth-death (Bd) update rule has two steps. First, a node is randomly selected out of the total population, with probability proportional to its fitness. Second, a neighbor of the first node is chosen, uniformly at random, and takes on the type of the first node.

In a complete graph, all nodes are adjacent. Therefore, the probability of adding a new invader, given there are currently $m$ invaders, is

$$p_m := P(\text{Choose an invader}) \cdot P(\text{Neighbor is resident}) = \frac{mr}{mr + (N-m)} \cdot \frac{N-m}{N-1}.$$

In the limit of infinite fitness, $(r \to \infty)$, the first term approaches one and we get

$$p_m := \frac{N-m}{N-1},$$

and the probability of the invader population ever decreasing is 0. So the time $T$ to invader fixation is sum of all the transition times $m \to m+1$ for $m = 1, 2, \ldots, N-1$. These transition times can be calculated as follows. For the population to take $t$ steps to go from $m$ to $m+1$ invaders, nothing must have happened for $t-1$ steps before advancing on the $t$'th step. The probability of this happening is exactly

$$p_m(1 - p_m)^{t-1}.$$

In other words, the time to add a new invader is exactly a geometric random variable. Therefore, the total fixation time is just

$$T = \sum_{m=1}^{N-1} \text{Geo}(p_m) = \sum_{k=1}^{N-1} \text{Geo}\left(\frac{k}{N-1}\right).$$

This random variable $T$ describes a process identical to that of the coupon collector's problem (*Pósfai, 2010*; *Feller, 1968*). In both, we have a collection of $N-1$ nodes, and draw a random one with replacement at each time step. If we pick a healthy node, we relabel it and toss it back, and repeat until there are no healthy nodes left. By adapting classic results (*Erdős and Rényi, 1961*; *Baum and Billingsley, 1965*), we show in the Appendix that it is straightforward to find the

asymptotic distribution of $T$ as $N$ gets large. To normalize this distribution, note that its mean is $\mu = \sum_m p_m^{-1} \approx N\log(N) + N\gamma$. Then we find

$$\frac{T - \mu}{N} \xrightarrow{d} \text{Gumbel}(-\gamma, 1). \tag{2}$$

Here $\gamma \approx 0.5772$ is the Euler-Mascheroni constant, $\xrightarrow{d}$ denotes convergence in distribution, and a Gumbel$(\alpha, \beta)$ random variable has a density given by

$$h(x) = \beta^{-1} e^{-(x-\alpha)/\beta} \exp\left(-e^{-(x-\alpha)/\beta}\right). \tag{3}$$

This prediction for the normalized distribution of the incubation period $T$ agrees with simulations on large networks (*Figure 6a*).

A Gumbel distribution of incubation periods has previously been obtained for a variant of this model. Instead of working with the large-$N$ limit of a complete graph, it assumed a continuous-time birth-death model of an invading microbial population whose dynamics were governed by differential equations (*Williams, 1965*).

## Birth-death, other solvable networks

The analysis of the finite-$N$ complete graph sets up an important framework that can be applied to more complicated networks. For example, in the Appendix we prove that the distribution of fixation times $T$ for a star network also converges to a Gumbel for $N \gg 1$, specifically:

$$\frac{T - N^2\log(N) - (\gamma - 1)N^2}{N^2} \xrightarrow{d} \text{Gumbel}(-\gamma, 1). \tag{4}$$

This prediction matches simulations (*Figure 6b*).

The same framework also applies to a one-dimensional (1D) ring lattice, but instead of using the coupon-collector framework, we need to cite the Lindeberg-Feller central limit theorem (*Durrett, 1991*). As shown in the Appendix, this gives us

$$\frac{T - (N^2 - N)/2}{(2N^3 - 3N^2 + N)/6} \xrightarrow{d} \text{Normal}(0, 1). \tag{5}$$

This prediction agrees with simulations (*Figure 6c*).

For a two-dimensional square lattice, it is more difficult to produce analytical results that are both rigorous and exact. But by making an approximation based on the geometry of the lattice, and using the fact that the population growth rate is proportional to its surface area (see the Appendix, "Normally distributed fixation times for 2D lattice'), we can make a non-rigorous analytical guess about the distribution of the fixation times $T$. Via these arguments, and given $\mu = \mathrm{E}[T]$ and $\sigma^2 = \mathrm{Var}(T)$, we predict

$$\frac{T - \mu}{\sigma} \xrightarrow{d} \text{Normal}(0, 1). \tag{6}$$

Despite the approximation, this prediction works well (*Figure 6d*).

By similar arguments, we predict that lattices of dimension $d \geq 3$ have right-skewed asymptotic distributions of fixation times. Specifically, given $\eta := 1 - 1/d$, we predict

$$\text{Skew}(T) := \frac{E[(T - \mu)^3]}{\sigma^3} = \frac{2\zeta(3\eta)}{\zeta(2\eta)^{3/2}}, \tag{7}$$

where $\zeta$ is the Riemann zeta function. The methods used to derive that can also be used to create approximate finite-size distributions for the lattices (*Figure 6e*).

In particular, we predict positive skew for all $d \geq 3$ and for the skew to increase monotonically with dimension (see the Appendix). Meanwhile, both 1D and 2D lattices have normal asymptotic distributions, and therefore no skew. This establishes $d = 2$ as a critical dimension in these dynamics, transitioning from zero skew to positive skew.

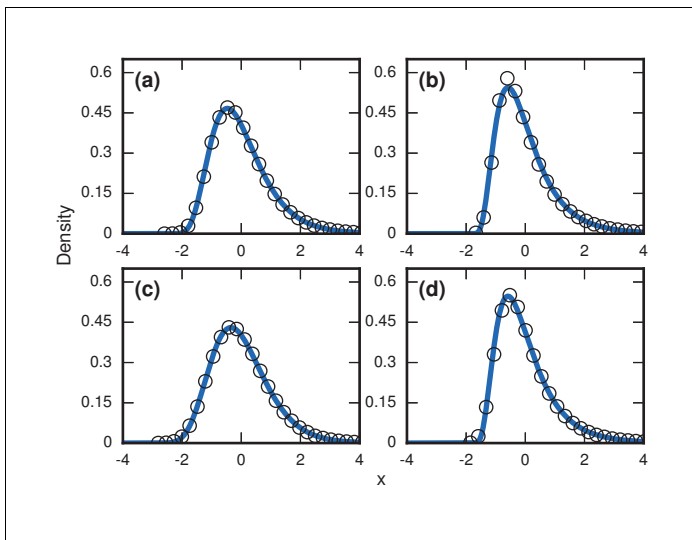

**Figure 9.** Robustness to population variability. Simulated, fitted, and normalized distributions of incubation periods for Birth-death dynamics on a complete graph that initially has $N = 500$ nodes. Invader fitness is set at $r = 10$. The blue curves indicate noncentral lognormals fitted via the method of moments. (**a**) Constant total population. (**b**) Growing population. At every time step, there is a constant $1/N$ chance that a new resident node will appear. The new node is adjacent to all preexisting nodes. (**c**) Shrinking population. At every time step, there is a constant $1/N$ chance that a random resident node will be removed. (**d**) Randomly varying population. At every time step, a resident node is either added or removed from the population, both events occurring with probability $1/2$.

DOI: https://doi.org/10.7554/eLife.30212.014

Incidentally, these arguments also suggest that appropriate infinite-dimensional networks will asymptotically have a Gumbel distribution. This is numerically true for the Erdős-Rényi random graph (*Figure 6f*).

For more complex networks, such as the Watts-Strogatz small-world network, the $k$-regular random graph, and the Barabasi-Albert scale-free network, we currently lack theory to predict the asymptotic distributions analytically. However, numerical simulations produce simulations that are all well-approximated by a noncentral lognormal, obeying Sartwell's law (*Sartwell, 1950*) (*Figure 7a,c, e*).

*Table 1* shows that geometric standard deviations of the incubation period distributions for *all* of these networks fall around $1.1 - 1.4$, in agreement with the dispersion factors of $1.1 - 1.5$ observed for many infectious diseases (*Sartwell, 1950*; *Horner and Samsa, 1992*).

## Random walk skewness

So far we have focused on infinitely fit invaders ($r \to \infty$). Now we consider the opposite extreme, where invaders have nearly neutral fitness ($r \approx 1$) relative to the residents. We will show that right-skewed distributions of incubation periods occur in this limit as well, but for a completely different reason than coupon collection.

The analysis is again simplest for the complete graph, so we return to that case. As before, the probability of an invader replacing a resident in the next time step is

$$p_m^+ := \frac{mr}{mr + (N - m)} \cdot \frac{N - m}{N - 1}.$$

Similarly, the probability of an invader being replaced by a resident in the next time step is

$$p_m^- := \frac{N - m}{mr + (N - m)} \cdot \frac{m}{N - 1}.$$

So the probability of the next replacement adding a new invader is

$$q := \frac{p_m^+}{p_m^+ + p_m^-} = \frac{r}{r+1}.$$

This defines a random walk with drift $q$ on the invader population.

Only a special subset of these walks are relevant to the computation of the incubation period distribution. For the incubation period to be well-defined, the invader population must not go extinct. Therefore, we need to condition on the fact that the invader population $m$ hits $N$ before it ever hits 0. For the limiting case $r = 1$, corresponding to a perfectly neutral invader, we can show with martingale methods that the resulting distribution of incubation periods will be strongly skewed to the right as $N$ gets large (see the Appendix). This is to be expected: there are only a few ways to walk from one to $N$ quickly, while there are many ways to have a long, meandering excursion before finally getting there.

The variance from this conditioned random walk process tends to drown out the effects of network topology. The distribution of incubation periods ends up looking similar for diverse networks (Figure 8), including complex networks (Figure 7b,d,f). So even though no coupon collection happens at low finesses $r \approx 1$, the effect of the conditioned random walk is more than enough to generate right-skewed distributions of incubation periods. In fact, this conditioned random walk mechanism at low $r$ produces an even higher dispersion factor ($\approx 1.7$) than coupon collection does at high $r$ (see Table 1).

## Influence of non-static population

In many diseases, it is unlikely that the total network size would remain constant in time. For example, targeted radiation and chemotherapy leads to a loss of mass in both the tumor and the substrate tissue. Depending on the specific physical case, the population levels of invaders and residents can have many nontrivial time dependencies. As a first-order examination of the effects of time-varying populations, three simple cases were considered on the complete graph for the intermediate fitness of $r = 10$. As a baseline, the distribution for a constant population was measured in Figure 9a.

We considered a case when the resident population was growing. At every time step, a new resident node was added with probability $1/N$, which was chosen so that takeover would happen in finite time. Even still, the majority of the run will be spent when the resident population is small, with takeovers and new additions occurring at a roughly even pace. This led to an accentuated level of right skew in Figure 9b.

We then considered a case where the resident population was constantly shrinking. Again, the probability of change was $1/N$ every time step, but this time it decreased the resident population by 1. While there is still a visible right skew in Figure 9c, it was somewhat lessened due to the global shrinkage speeding up the coupon collecting process.

Finally, we considered a randomly varying resident population. Here, the resident population increases or decreases by one every time step, each with probability 1/2. This random-walking population level also leads to an extreme level of skew in Figure 9d.

## Acknowledgements

We thank David Aldous, Rick Durrett, Remco van der Hofstad, Lionel Levine, Piet Van Mieghem, and Steve Schiff for comments.

## Additional information

### Funding

| Funder | Grant reference number | Author |
| --- | --- | --- |
| National Science Foundation | DMS-1513179 | Steven H Strogatz |
| National Institutes of Health | Loan Repayment Grant | Jacob G Scott |
| National Science Foundation | Graduate Student Fellowship DGE-1650441 | Bertrand Ottino-Loffler |

| National Science Foundation | CCF-1522054 | Steven H Strogatz |

The funders had no role in study design, data collection and interpretation, or the decision to submit the work for publication.

## Author contributions

Bertrand Ottino-Loffler, Conceptualization, Resources, Supervision, Funding acquisition, Investigation, Project administration, Writing—review and editing, Provided mathematical background, designed the study and wrote the manuscript; Jacob G Scott, Conceptualization, Software, Formal analysis, Investigation, Visualization, Methodology, Writing—original draft, Writing—review and editing, Provided mathematical analysis and simulations, designed the study and wrote the manuscript; Steven H Strogatz, Conceptualization, Supervision, Investigation, Methodology, Writing—review and editing, Provided biomedical background, designed the study and wrote the manuscript

## Author ORCIDs

Bertrand Ottino-Loffler (iD) http://orcid.org/0000-0001-6839-5510
Jacob G Scott (iD) http://orcid.org/0000-0003-2971-7673
Steven H Strogatz (iD) http://orcid.org/0000-0003-2923-3118

## Decision letter and Author response

Decision letter https://doi.org/10.7554/eLife.30212.019
Author response https://doi.org/10.7554/eLife.30212.020

# Additional files

## Supplementary files

• Source code 1. 'FigureDataScripts.zip': Python scripts and documentation to produce datasets used in *Figures 3–9*.
DOI: https://doi.org/10.7554/eLife.30212.015

• Transparent reporting form
DOI: https://doi.org/10.7554/eLife.30212.016

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

# Appendix 1

DOI: https://doi.org/10.7554/eLife.30212.017

## Agreement of geometric and exponential variables I

Proposition: Suppose we have a family of sequences $(p_m)_{m=1}^{M}$, with $0 \leq p_m \leq 1$ for all $m$ and $M$, where $p_m$ may depend on $M$. Define $\mathrm{Geo}(p)$ to be a geometric random variable with distribution

$$P(\mathrm{Geo}(p) = k) = (1-p)^{k-1} p$$

for $k = 1, 2, \ldots$. Further, let $\mathcal{E}(p)$ be an exponential random variable with distribution

$$P(\mathcal{E}(p) = x)dx = pe^{-px}dx$$

for $x \geq 0$. Given some function $L := L(M)$ such that $\lim_{M \to \infty} L = \infty$ and

$$\lim_{M \to \infty} \sum_{m=1}^{M} \frac{1}{p_m L^2} = 0, \tag{8}$$

and given $T_G := \sum_{m=1}^{M} X(p_m)$, $T_E := \sum_{m=1}^{M} \mathcal{E}(p_m)$, and $\mu := \sum_{m=1}^{M} 1/p_m$, then

$$\frac{T_G - \mu}{L} \sim \frac{T_E - \mu}{L}. \tag{9}$$

The symbol "$\sim$" means the ratio of characteristic functions goes to one as $N$ gets large. That is, the random variables on both sides converge to each other in distribution as $M$ gets large.

Proof: The proof of this claim simply involves calculating the characteristic functions and taking a limit. We have presented the details elsewhere (**Ottino-Löffler et al., 2017**).

## Agreement of geometric and exponential variables II

Proposition: Given the setup in the previous proposition, define $\sigma_G^2 = \mathrm{Var}(T_G)$ and $\sigma_E^2 = \mathrm{Var}(T_E)$. If

$$\lim_{M \to \infty} \frac{\sum_{m=1}^{M} p_m^{-1}}{\sum_{m=1}^{M} p_m^{-2}} = 0, \tag{10}$$

then

$$\frac{T_G - \mu}{\sigma_G} \sim \frac{T_E - \mu}{\sigma_E}. \tag{11}$$

Proof: Our first goal is to show that

$$\frac{T_G - \mu}{\sigma_G} \sim \frac{T_E - \mu}{\sigma_G}.$$

We do this by using the proposition in 'Agreement of geometric and exponential variables I', substituting $\sigma_G$ for $L$. First we check that **Equation (8)** is satisfied. Notice that

$$\begin{aligned}
\lim_{M \to \infty} \sum_{m=1}^{M} \frac{1}{p_m \sigma_G^2} &= \lim_{M \to \infty} \frac{\sum_{m=1}^{M} p_m^{-1}}{\sum_{m=1}^{M} p_m^{-2} - p_m^{-1}} \\
&= \lim_{M \to \infty} \frac{\sum_{m=1}^{M} p_m^{-1} / \sum_{m=1}^{M} p_m^{-2}}{1 - \sum_{m=1}^{M} p_m^{-1} / \sum_{m=1}^{M} p_m^{-2}} \\
&= \frac{0}{1-0} = 0
\end{aligned}$$

by hypothesis. Hence

$$\frac{T_G - \mu}{\sigma_G} \sim \frac{T_E - \mu}{\sigma_G} = \frac{\sigma_E}{\sigma_G} \frac{T_E - \mu}{\sigma_E}.$$

But notice that

$$
\begin{aligned}
\lim_{M \to \infty} \frac{\sigma_E^2}{\sigma_G^2} &= \lim_{M \to \infty} \frac{\sum_{m=1}^{M} p_m^{-2}}{\sum_{m=1}^{M} p_m^{-2} - p_m^{-1}} \\
&= \lim_{M \to \infty} \frac{1}{1 - \sum_{m=1}^{M} p_m^{-1} / \sum_{m=1}^{M} p_m^{-2}} \\
&= 1.
\end{aligned}
$$

Therefore, the proposition is proven.

## Condition for normality

Proposition: Let $T = \sum_{m=1}^{M} \mathcal{E}(p_m)$, define $\sigma^2 = \mathrm{Var}(T) = \sum_m p_m^{-2}$, and let $\lim_{M \to \infty} p_m \sigma = \infty$. If

$$\lim_{M \to \infty} \sum_{m=1}^{M} \exp(-\epsilon p_m \sigma) = 0, \tag{12}$$

then

$$\frac{T - \mu}{\sigma} \xrightarrow{d} \mathrm{Normal}(0, 1). \tag{13}$$

Proof: Apply the Lindeberg-Feller central limit theorem (**Durrett, 1991**) to the random variables

$$Y_{m,M} := \frac{\mathcal{E}(p_m) - 1/p_m}{\sigma}.$$

By construction, $\sum_m (Y_m, M) = (T - \mu)/\sigma$, $E[Y_{m,M}] = 0$ and $\sum_m E[Y_{m,M}^2] = 1$. So in order to apply the theorem (and thereby get our desired result), we simply need satisfy the Lindeberg condition for any $\epsilon > 0$, as given by

$$\lim_{M \to \infty} \mathrm{Lind}_M := \lim_{M \to \infty} \sum_{m=1}^{M} E[Y_{m,M}^2; |Y_{m,M}| > \epsilon] = 0.$$

Notice that $Y_{m,M} < -\epsilon$ implies

$$\mathcal{E}(p_m) < p_m^{-1} - \epsilon \sigma_E^2 = p_m^{-1}(1 - \epsilon p_m \sigma).$$

By hypothesis, the right hand side will eventually be less than 0, meaning that eventually $Y_{m,M} < -\epsilon$ will be impossible. So if we define $c_m = 1 + \epsilon p_m \sigma$, then we know that

$$\lim_{M \to \infty} \mathrm{Lind}_M = \lim_{M \to \infty} \sum_{m=1}^{M} E[Y_{m,M}^2; Y_{m,M} > \epsilon].$$

So for large enough $M$, we have

$$
\begin{aligned}
\mathrm{Lind}_M \\
&:= \sum_{m=1}^{M} \int_{c_m/p_m}^{\infty} \left( \frac{x - 1/p_m}{\sigma} \right)^2 e^{-p_m x} p_m \, dx \\
&= \sum_{m=1}^{M} \frac{1}{\sigma^2 p_m^2} \int_{c_m}^{\infty} (y - 1)^2 e^{-y} \, dy \\
&= \sum_{m=1}^{M} \frac{1}{\sigma^2 p_m^2} e^{-c_m} (c_m^2 + 1) \\
&= \sum_{m=1}^{M} \frac{1}{e \sigma^2 p_m^2} e^{-\epsilon p_m \sigma} (2 + 2\epsilon p_m \sigma + \epsilon^2 p_m^2 \sigma^2).
\end{aligned}
$$

By hypothesis, $p_m\sigma$ grows without bound, so the term $p_m^2\sigma^2$ will be dominant. Therefore, there is some constant $D$ such that we can create the upper bound

$$\mathrm{Lind}_M \leq \sum_{m=1}^{M} \frac{1}{\sigma^2 p_m^2} e^{-\epsilon p_m \sigma} D\sigma^2 p_m^2$$

$$\leq D \sum_{m=1}^{M} \exp(-\epsilon p_m \sigma).$$

From here, we can apply the hypothesis to get $\lim_{M\to\infty} \mathrm{Lind}_M \leq \lim_{M\to\infty} D\sum_{m=1}^{M} \exp(-\epsilon p_m \sigma) = 0.$

Thus the Lindeberg condition holds. Therefore, by applying the theorem, we conclude that

$$\sum_m (Y_{m,M}) = \frac{T-\mu}{\sigma} \xrightarrow{d} \mathrm{Normal}(0,1).$$

## Normally distributed fixation times for 1D lattice

Let us start with a one-dimensional (1D) ring of $N$ nodes, with exactly one invader at the start. Under infinite-$r$ Birth-death (Bd) dynamics, a uniform random invader gives birth at every time step and replaces one of its neighbors, also uniformly at random.

Because of the simple topology of the ring, the growing chain of invaders will always advance from the left or right ends. This means that if there are currently $m$ invaders, then the probability of a new invader being added in this time step is exactly given by

$$p_m = \frac{1}{m}$$

for $m = 1, 2, \ldots, M$, where we have defined $M := N - 1$.

The probability of spending exactly $t$ time steps at $m$ invaders is given by the odds of doing nothing for exactly $t-1$ steps and then advancing at the last step, and so is given by $(1-p_m)^{t-1} p_m$. In other words, the time spent with $m$ invaders is given by the geometric random variable $\mathrm{Geo}(p_m)$. Therefore the total fixation time $T$ is given by $T = \sum_{m=1}^{M} \mathrm{Geo}(p_m)$.

By applying the results of section 'Agreement of geometric and exponential variables II', we can switch to using exponential random variables. From here we wish to apply results from section 'Condition for normality', so we need to check if $\sum_{m=1}^{M} \exp(-\epsilon p_m \sigma)$ converges to zero, given $\sigma = \sum_{m=1}^{M} p_m^{-2}$. Using asymptotics and a constant $D$, we find

$$\sum_{m=1}^{M} \exp(-\epsilon p_m \sigma) \leq \sum_{m=1}^{M} \exp\left(\frac{-D}{m} M^{3/2}\right)$$

$$\leq \sum_{m=1}^{M} \exp\left(\frac{-D}{M} M^{3/2}\right)$$

$$= M \exp\left(-D\sqrt{M}\right) \to 0.$$

This lets us cite our proposition, and conclude that the fixation time is asymptotically distributed as a normal, with

$$\sum_m (Y_{m,M}) = \frac{T-\mu}{\sigma} \xrightarrow{d} \mathrm{Normal}(0,1).$$

## Gumbel distributed fixation times for star graph

Next consider the infinite-$r$ Bd dynamics on a star graph. This network consists of one 'hub' node and $N$ 'spoke' nodes, with edges exclusively between the hub and spokes. We will place the initial invader at the hub, since starting from a spoke is a trivial perturbation off of that.

Now, given that $m$ of the $N$ spokes are invaders, then the odds of another one turning into an invader in one time step is simply the odds of choosing the hub from the set of all invaders, times the probability of replacing an existing healthy spoke. So

$$p_m = \frac{1}{m+1} \cdot \frac{N-m}{N}$$

for $m = 0, 1, \ldots, N-1$. As before, the fixation time $T$ is the sum of geometric random variables $\mathrm{Geo}(p_m)$. However, it is easy to use section 'Agreement of geometric and exponential variables II' to show that the total fixation time is well approximated by the sum of exponential random variables $\mathcal{E}(p_m)$, given we normalize by $N^2$. So we know

$$\frac{T - E(T)}{N^2} = \sum_{m=1}^{N-1} \frac{\mathrm{Geo}(p_m) - 1/p_m}{N^2} \sim \sum_{m=1}^{N-1} \frac{\mathcal{E}(p_m) - 1/p_m}{N^2}. \tag{14}$$

The crux of finding the limiting distribution of $T$ is noticing that $Np_m \approx (N-m)/N$ for large $m$. The sequence $r_m := (N-m)/N, m = 0, \ldots, N-1$ corresponds to the well-known *coupon collector's problem* (see main text). Imagine a child trying to complete a collection of $N$ cards by buying one random card each week. The probability of getting a new card the first week is 1; the probability of getting a new card after the first has been collected is $(N-1)/N$; the probability of getting another new card after two cards have been collected is $(N-2)/N$; and so on until the probability of getting the last card is $1/N$.

The time $T_C$ to complete this collection (which is also well approximated by the sum of exponentials) has been the subject of much historical study. In fact, an exact distribution for $T_C$ in the limit of large $N$ is known (**Ottino-Löffler et al., 2017**; **Pósfai, 2010**; **Feller, 1968**; **Erdős and Rényi, 1961**; **Baum and Billingsley, 1965**; **Rubin and Zidek, 1965**), and is given in normalized form by

$$\frac{T_C - E(T_C)}{N} \sim \sum_{m=1}^{N-1} \frac{\mathcal{E}(r_m) - 1/r_m}{N} \xrightarrow{d} \mathrm{Gumbel}(-\gamma, 1), \tag{15}$$

where $\gamma \approx 0.5772$ is the Euler-Mascheroni constant.

Therefore, if we can connect our fixation time $T$ to the coupon collector's time $T_C$, we would know its distribution. We do so by taking the ratio of their respective characteristic functions, using their respective approximations as exponential random variables. Letting $k = N - m$, we find a characteristic function

$$\phi :=$$
$$= E\left[\exp\left(it \sum_{k=1}^{N} \frac{\mathcal{E}(p_k) - 1/p_k}{N^2}\right)\right]$$
$$= \prod_{k=1}^{N} \frac{\exp -it/(N^2 p_k)}{1 - it/(N^2 p_k)}$$

for our fixation time, and

$$\phi_C :=$$
$$= E\left[\exp\left(it \sum_{k=1}^{N} \frac{\mathcal{E}(r_k) - 1/r_k}{N}\right)\right]$$
$$= \prod_{k=1}^{N} \frac{\exp -it/k}{1 - it/k}$$

for the coupon collector's time.

Taking the ratio gives

$$\frac{\phi_C}{\phi} = \prod_{k=1}^{N} \exp\left[\frac{-it}{N}\left(1 - \frac{1}{k}\right) + \log\left(1 + \frac{it\, k - 1}{N\, k - it}\right)\right].$$

Taking the Taylor expansion of the logarithm for $N \gg 1$, we can substitute in an appropriate function $R_m$ which gets small as $N$ gets large. Thus

$$\frac{\phi_C}{\phi} = \exp\left[\frac{it}{N}\sum_{k=1}^{N}\frac{it(k-1)}{k(k-it)} + \frac{t^2}{2}\sum_{k=1}^{N}\frac{R_m}{N^2}\left(\frac{k-1}{k-it}\right)^2\right].$$

The first sum is bounded above in norm by a constant times $\log(N)$, so the first term goes to zero as $N$ gets large. Similarly, the second term goes to zero quickly, meaning that $\phi_C/\phi \to 0$. Hence

$$\sum_{k=1}^{N}\frac{\mathcal{E}(p_k) - 1/p_k}{N^2} \sim \sum_{k=1}^{N}\frac{\mathcal{E}(r_k) - 1/r_k}{N}. \tag{16}$$

Using *Equation (16)* to connect *Equation (14)* to *Equation (15)*, we find

$$\frac{T - E(T)}{N^2} \xrightarrow{d} \text{Gumbel}(-\gamma, 1), \tag{17}$$

as desired.

## Normally distributed fixation times for 2D lattice

We wish to find the limiting distribution of fixation times of infinite-$r$ Bd growth on a $d$-dimensional square lattice, assuming periodic boundaries. We will eventually focus on the two-dimensional (2D) lattice, but let us set up every case for $2 \le d < \infty$ right now.

Unlike the previous cases, the probability of adding a new invader always depends on the exact configuration of the existing invaders. So we can no longer define exact values for $p_m$ that describe the fixation time $T$ as a simple sum of random variables.

But even though we do not know the exact shape of the invader cluster, the simple network structure can motivate a reasonable approximation. In particular, given a sufficiently smooth and convex volume $V$ in a $d$-dimensional lattice, we should expect the volume to have a surface area proportional to $V^\eta$, where $\eta = 1 - 1/d$.

Assuming this to be true, recall the basic dynamics of infinite-$r$ Bd with $N$ nodes and $m$ invaders. First, we uniformly select one node out of the population of invaders, which is always a probability of $1/m$ per node. Then we replace one of the invader's neighbors, uniformly at random. However, only invaders on the surface of the cluster even have a chance at replacing a healthy node!

Given sufficient regularity of the boundary of the cluster, this means that the probability of an invader replacing a healthy node is proportional to

$$q_m = \frac{1}{m} \cdot \text{Surface area of the invader cluster.}$$

Using this logic, we expect the probability of adding an invader to go as $m^\eta/m$ at the start. However, remember we have a periodic lattice, so using $m^\eta$ as the surface area of the cluster stops being true halfway through, and using the smaller healthy cluster's volume is a better approximation. In other words, the surface area grows as $m^\eta$ at the start when the cluster begins forming, and shrinks as $(N-m)^\eta$ at the end when there are only a few healthy cells left.

This intuitive reasoning suggests that the 'true' probability of adding a new invader, given that there are already $m$ of them, should be roughly proportional to

$$q_m = \frac{\min(m, N-m)^\eta}{m}. \tag{18}$$

The fact that we only estimated $q_m$ up to proportionality is adequate, because when we use section 'Agreement of geometric and exponential variables II', any multiplicative factors will just be absorbed by the variance anyway when we write down the normalized distribution.

The case of $d = 2$ is special, so let us plug in $d = 2$ and $\eta = 1/2$ where appropriate. This gives

$$q_m = \frac{\min(m, N-m)^{1/2}}{m}.$$

Now use section 'Agreement of geometric and exponential variables II'. Skip ahead to defining $T$ to be the sum of exponential random variables, and split the sum in half. Thus

$$T := T_a + T_b := \sum_{m=1}^{N/2-1} \mathcal{E}(q_m) + \sum_{m=N/2}^{N-1} \mathcal{E}(q_m).$$

We assume $N$ is even without loss of generality.

First, we show that $T_a$ is normally distributed using the section 'Condition for normality'. To use this result, we first need to calculate $\mathrm{Var}(T_a)$. This is exactly

$$\mathrm{Var}(T_a) = \sum_{m=1}^{N/2-1} q_m^{-2} = \sum_{m=1}^{N/2-1} m = \frac{(N/2-1)^2 + (N/2-1)}{2}.$$

Therefore, we have

$$q_m^2 \mathrm{Var}(T_a) = \frac{1}{m} \frac{(N/2-1)^2 + (N/2-1)}{2} \geq \frac{1}{m} \frac{N^2}{8} \geq \frac{1}{N} \frac{N^2}{8} = \frac{N}{8} \to \infty$$

as $N \to \infty$, so the first condition is satisfied.

Next, we need to show that a certain sum of exponentials converges to zero. In particular, for any $\epsilon > 0$, we examine

$$S_N := \sum_{m=1}^{N/2-1} \exp\left(-\epsilon q_m \sqrt{\mathrm{Var}(T_a)}\right).$$

We can make the bound $\sqrt{\mathrm{Var}(T_a)} > N/8$ and $q_m = 1/\sqrt{m} > 1/\sqrt{N}$, so

$$S_N \leq \sum_{m=1}^{N/2-1} \exp\left(-\epsilon \sqrt{N}/8\right) \leq N \exp\left(-\epsilon \sqrt{N}/8\right).$$

Therefore $S_N \to 0$ as $N$ gets large. So the second condition is satisfied. Therefore, $T_a$ is distributed according to a normal.

However, we need to do the same to $T_b$. So let us estimate the variance of this second contribution. Letting $k = N - m$, we get

$$
\begin{aligned}
\mathrm{Var}(T_b) &= \sum_{k=1}^{N/2} q_{N-k}^{-2} = \sum_{k=1}^{N/2} \left(\frac{N-m}{\sqrt{k}}\right)^2 \\
&= N^2 \left(\sum_{k=1}^{N/2} \frac{1}{k} - \frac{2}{N} \sum_{k=1}^{N/2} 1 + \frac{1}{N^2} \sum_{k=1}^{N/2} k\right) \\
&= N^2 \left(\sum_{k=1}^{N/2} \frac{1}{k} - 1 + \frac{1}{8} + \frac{1}{8N}\right).
\end{aligned}
$$

So for large $N$, we obtain the following bound:

$$\mathrm{Var}(T_b) \geq \frac{N^2}{4} \log(N).$$

Therefore,

$$q_k^2 \mathrm{Var}(T_b) \geq \frac{k}{(N-k)^2} \frac{N^2}{4} \log(N) \geq \frac{1}{4} \log N.$$

This satisfies the first condition from section 'Condition for normality'.

To satisfy the second condition, we again fix some arbitrary $\epsilon > 0$ and calculate a certain sum of exponentials. By calculating and choosing careful bounds, we get

$$
\begin{aligned}
S_N \quad &:= \sum_{k=1}^{N/2} \exp\left(-\epsilon q_k \sqrt{\mathrm{Var}(T_b)}\right) \\
&\leq \sum_{k=1}^{N} \exp\left(-\frac{\epsilon}{2}\frac{\sqrt{k}}{N-k}N\sqrt{\log(N)}\right) \\
&\leq \sum_{k=1}^{N} \exp\left(-\frac{\epsilon}{2}\sqrt{k\log(N)}\right)
\end{aligned}
$$

This sum can be approximated from above by an appropriate integral:

$$
\begin{aligned}
S_N \quad &\leq \int_0^\infty \exp\left(-\frac{\epsilon}{2}\sqrt{x\log(N)}\right)dx \\
&= \frac{8}{\epsilon^2 \log(N)}.
\end{aligned}
$$

Therefore, $S_N \to 0$ as $N$ gets large. This satisfies the second condition in section 'Condition for normality', meaning that we now know that $T_b$ is distributed as a normal.

Since the sum of normal variables returns a normal variable, this means that $T = T_a + T_b$ is also normal. Hence, we expect for the fixation time of infinite-$r$ Bd on a 2D lattice to be distributed as a normal, like the 1D ring lattice but, as we will now show, unlike $d \geq 3$.

## Non-normality for $d \geq 3$

Here, we wish to find the limiting distribution of fixation times of infinite-$r$ Bd growth on a $d$-dimensional square lattice, assuming periodic boundaries. Right now, we will look only at $3 \leq d < \infty$, since $d = 1, 2,$ and $\infty$ are special cases.

We did the bulk of the setup for this case in section 'Normally distributed fixation times for 2D lattice', so we have the approximate probabilities of adding an invader to be given by

$$
q_m = \frac{\min(m, N-m)^\eta}{m}
$$

as before. And again, we define the 'approximate' fixation time $T$ to be the sum of the exponential random variables $\mathcal{E}(q_m)$. Splitting the sum into a front and back half gives

$$
T := T_a + T_b := \sum_{m=1}^{N/2-1} \mathcal{E}(q_m) + \sum_{m=N/2}^{N-1} \mathcal{E}(q_m).
$$

But even with such aggressive approximations, we cannot present a closed form for the distribution of $T$. However, we can still calculate an important quantity: the skew of the distribution.

Since we use section 'Agreement of geometric and exponential variables II', let us skip to defining $T$ to be the sum of exponential random variables, and split the sum in half, as we did with the 2D lattice. Thus

$$
T := T_a + T_b := \sum_{m=1}^{N/2-1} \mathcal{E}(q_m) + \sum_{m=N/2}^{N-1} \mathcal{E}(q_m).
$$

We assume $N$ is even without loss of generality.

First, let us find which half contributes more variance. Bound $\mathrm{Var}(T_a)$ as

$$\mathrm{Var}(T_a) \;=\; \sum_{m=1}^{N/2-1} q_m^{-2} = \sum_{m=1}^{N/2-1} m^{2/d}$$

$$\leq \int_0^{N/2} x^{2/d}\,dx \leq N^{2/d+1}.$$

We similarly approximate $\mathrm{Var}(T_b)$, setting $k = N - m$ and finding

$$\mathrm{Var}(T_b) \;=\; \sum_{k=1}^{N/2} q_{N-k}^{-2} = \sum_{k=1}^{N/2} \frac{(N-k)^2}{k^{2\eta}}$$

$$= N^2 \left( \sum_{k=1}^{N/2} \frac{1}{k^{2\eta}} - \frac{2}{N}\sum_{k=1}^{N/2} k^{1-2\eta} + \frac{1}{N^2}\sum_{k=1}^{N/2} k^{2/d} \right).$$

Since $\eta \geq 2/3$, only the first term survives as $N$ gets large, so

$$\mathrm{Var}(T_b) \to N^2 \zeta(2\eta) \tag{19}$$

where $\zeta$ is the usual Riemann zeta function. Notice that $2 > 2/d + 1$ for $d \geq 3$.

To use both these variances, recall the skewness summation formulation: if we have random variables $X_i$ with variances $\sigma_i^2$ and skews $\kappa_i$, then their sum has a skewness of

$$\mathrm{Skew}\left(\sum_i X_i\right) = \frac{\sum_i \kappa_i \sigma_i^3}{\left(\sum_i \sigma_i^2\right)^{3/2}}. \tag{20}$$

So this means that

$$\mathrm{Skew}(T) \;=\; \frac{\mathrm{Skew}(T_a)\mathrm{Var}(T_a)^{3/2} + \mathrm{Skew}(T_b)\mathrm{Var}(T_b)^{3/2}}{\left(\mathrm{Var}(T_a) + \mathrm{Var}(T_b)\right)^{3/2}}$$

$$= \frac{\mathrm{Skew}(T_a)(\mathrm{Var}(T_a)/\mathrm{Var}(T_b))^{3/2} + \mathrm{Skew}(T_b)}{\left(1 + \mathrm{Var}(T_a)/\mathrm{Var}(T_b)\right)^{3/2}}$$

$$\to \mathrm{Skew}(T_b)$$

as $N \to \infty$. Here, we have used the fact that $T_a$ has a finite skew (actually, it is easy to use section 'Condition for normality' to show that $T_a$ is distributed as a normal, and thus has zero skew.) Hence the asymptotic skew of $T$ is just the asymptotic skew of $T_b$.

We can calculate the skew of $T_b$ by reusing **Equation (20)**, this time on the exponential variables defining $T_b$. Therefore

$$\mathrm{Skew}(T_b) = \frac{\sum_{k=1}^{N/2} 2 q_{N-k}^{-3}}{\left(\sum_{k=1}^{N/2} q_{N-k}^{-2}\right)^{3/2}}.$$

By **Equation (19)**, the denominator limits to $N^3 \zeta(2\eta)^{3/2}$. Meanwhile, the numerator looks like

$$2\sum_{k=1}^{N/2} \frac{1}{k^{3\eta}}(N-k)^3$$

$$= 2N^3\left( \sum_{k=1}^{N/2} \frac{1}{k^{3\eta}} - \frac{3}{N}\sum_{k=1}^{N/2} k^{1-3\eta} + \frac{3}{N^2}\sum_{k=1}^{N/2} k^{2-3\eta} + \frac{1}{N^3}\sum_{k=1}^{N/2} k^{3/d} \right).$$

The first term in the parentheses converges to $\zeta(3\eta)$, whereas the rest of the terms converge to 0. Combining the numerator and denominator gives us the conclusion that the skew for the full distribution is given by

$$\text{Skew}(T) = \frac{2\zeta(3\eta)}{\zeta(2\eta)^{3/2}}. \tag{21}$$

Recall that $\eta = 1 - 1/d$, so the denominator diverges for $d = 2$ and both then numerator and denominator diverge for $d = 1$. However, since this expression for $\text{Skew}(T)$ is monotone increasing in $\eta$ for $2/3 \le \eta < 1$, every dimension $d \ge 3$ will attain a unique skew, and therefore a unique limiting distribution. Moreover, if we take $d \to \infty$, then $\eta \to 1$, and therefore the skew becomes $12\sqrt{6}\zeta(3)/\pi^3$, which is exactly the skew for a Gumbel distribution, supporting our assertion that high-dimensional systems attain higher skews.

For the purposes of estimating the distributions at finite $N$, it is convenient to use the random variable

$$F(d) := \frac{\sum_{k=1}^{N} \mathcal{E}(p_k) - p_k^{-1}}{\sum_{k=1}^{N} p_k^{-2}}$$

where

$$p_k = \frac{k^{1-1/d}}{N-k}.$$

Since the second half of the dynamics contribute the majority of the variance, we should expect this to provide a reasonable approximation of $T$.

## Asymptotic skew of conditioned random walk

When $r = 1$ for Bd dynamics on the complete graph, the number of invaders becomes very flexible. In fact, the probability of adding an invader on any time step is exactly equal to the probability of removing an invader, so

$$p_m^+ = \frac{m(N-m)}{N(N-1)} = p_m^-.$$

Hence the probability that the next event increases the number of invaders is always 1/2. This means that, if we ignore the waiting times, the population of invaders obeys a simple random walk on the values $m = 1, ..., N - 1$ with 0 and $N$ acting as absorbing states. So we can understand the times required to take over the network by understanding these simple dynamics. To set up the analysis, let $X_n$ denote the number of invaders after $n$ population changes. Therefore

$$X_n = \sum_{i=1}^{n} x_i$$

where $x_i \in \{-1, +1\}$, each with probability 1/2. We will use the wait-omitted time $n$ in this section as a first-order approximation of the true takeover time. This way, the present analysis is generalizable to most networks. Moreover, scaling and numerical arguments based on the results here can show that the bulk of the final distribution is defined by this random-walk process.

Although we always start at $m = 1$, we only care about the invader takeover result because that is the only case for which disease symptoms would be manifested. Let us define the stopping time

$$S_m = \min\{n | X_n = m\},$$

which records the first time the random walk $X_n$ hits the value $m$. Given that the invader population cannot go negative or above $N$, the walk's stopping time is

$$S = \min(S_0, S_N).$$

In the main text, we cared only about the conditioned random walk times and whether they tended to be right skewed. So we now study the first few conditional moments

$$\mu_i := E(S^i|X_S = N),$$

for $i = 1, 2, 3$.

It isn't hard to set up a linear recurrence relation to find the probability of hitting 0 or $N$. In fact, if the state of the walk is at $m$, then the probability of hitting $N$ is exactly

$$P(X_S = N|X_n = m) = m/N.$$

This is a useful fact for simulation; instead of directly simulating $X_n$ and discarding all the cases that hit 0, we can directly simulate the conditioned random walk. If we define

$$Y_n = E(X_n|X_S = N),$$

and treat it as a Markov chain, then we can easily calculate the transition probabilities. By applying Bayes's law,

$$
\begin{aligned}
P(Y_n = m \to Y_{n+1} = m+1) &= \frac{P(X_n = m \to X_{n+1} = m+1 \text{ AND } X_S = N)}{P(X_S = N)} \\
&= \frac{1}{2}\frac{P(X_S = N|X_n = m+1)}{P(X_S = N|X_n = m)} \\
&= \frac{1}{2}\frac{m+1}{m}.
\end{aligned}
$$

While this speeds up simulations by a good deal in certain cases, it is not particularly useful for quantifying the distribution of the random walk times themselves.

To identify the moments of $T$, we will want to apply results from martingale theory in general, and optional stopping in particular. To start, define the random variable

$$M_n^{(1)} := X_n^3 - 3nX_n.$$

We are going to want this to be a martingale. Let's define $\mathcal{F}_n$ to be the sigma field consisting of all information from the first $n$ steps of the random walk. Therefore, $E(x_{n+1}|\mathcal{F}_n) = 0$, since we cannot predict the direction of the next step. However, we do know that $E(x_{n+1}^2|\mathcal{F}_n) = 1$, because the steps will always be size 1, regardless of our ignorance. Putting this together gives

$$
\begin{aligned}
E(M_{n+1}^{(1)}|\mathcal{F}_n) &= E\left(X_{n+1}^3 - 3(n+1)X_{n+1}|\mathcal{F}_n\right) \\
&= E\left((X_n + x_{n+1})^3 - 3(n+1)(X_n + x_{n+1})\right) \\
&= E\left(X_n^3 + 3x_{n+1}X_n^2 + 3x_{n+1}^2 X_n - 3(n+1)X_n - 3(n+1)x_{n+1}\right) \\
&= X_n^3 - 3nX_n \\
&= M_n^{(1)}.
\end{aligned}
$$

So $M_n^{(1)}$ well-approximates its future, meaning that it is a proper martingale.

Thank to this, we can cite an optional stopping theorem (**Durrett, 1991**). So we expect the expectation of this variable to be the same at the stopping time as at the start, so

$$E\left(M_0^{(1)}\right) = E\left(M_S^{(1)}\right). \tag{22}$$

Since we start at $n = 0$ and $X_0 = 1$, the left hand side trivially gives

$$E\left(M_0^{(1)}\right) = 1^3 - 3 \cdot 0 \cdot 1 = 1.$$

However, the right hand side gives something a bit more complicated, since we need to condition on the possible endpoints, remembering we start at $X_0 = 1$. So

$$
\begin{aligned}
E\left(M_S^{(1)}\right) &= P(X_S = N)E(M_S^{(1)}|X_S = N) + P(X_S = 0)E(M_S^{(1)}|X_S = 0) \\
&= \frac{1}{N}E\left(X_n^3 - 3nX_n|X_S = N\right) + \frac{N-1}{N}E\left(X_n^3 - 3nX_n|X_S = 0\right) \\
&= \frac{1}{N}\left(N^3 - 3\mu_1 N\right) + \frac{N-1}{N}E\left(0^3 - 3E(S|X_S = 0)0\right) \\
&= N^2 - 3\mu_1.
\end{aligned}
$$

Thus we have now calculated both sides of **Equation (22)**, meaning that we now have a value for the first moment of time of the conditioned random walk, given by

$$
\mu_1 = \frac{N^2 - 1}{3}.
$$

The procedure to find the next two moments is not too substantially different: just repeat the same steps of verification and evaluation on $M_n^{(1)}$'s siblings

$$
\begin{aligned}
M_n^{(2)} &= X_n^5 - 10nX_n^3 + (15n^2 + 10n)X_n \\
M_n^{(3)} &= X_n^7 - 21nX_n^5 + (105n^2 + 70n)X_n^3 - (105n^3 + 210n^2 + 112n)X_n.
\end{aligned}
$$

These two martingales reveal the next two moments, which are given by

$$
\begin{aligned}
\mu_2 &= \frac{7N^4 - 20N^2 + 13}{45} \\
\mu_3 &= \frac{31N^6 - 147N^4 + 189N^2 - 73}{315}.
\end{aligned}
$$

This is all we need to compute the asymptotic skew of the conditioned fixation times. In the limit of large $N$, only the dominant terms of each $\mu_i$ will survive. The $N$'s cancel out in this limit, leading to a constant given by

$$
\text{Skew} = \frac{\mu_3 - 3\mu_1\mu_2 + 2\mu_1^3}{\left(\mu_2 - \mu_1^2\right)^{3/2}} \approx \frac{8}{7}\left(\frac{5}{2}\right)^{1/2} \approx 1.807.
$$

The conclusion is that in the limit of large $N$, the skew of the distribution of fixation times for a conditioned random walk will always be positive. Therefore, we expect right-skewed distributions to be typical for the $r = 1$ limit of Birth-death dynamics.

