## [Decision Letter]

Thank you for submitting your article "Evolutionary dynamics of incubation periods" for consideration by *eLife*. Your article has been reviewed by three peer reviewers, and the evaluation has been overseen by a Reviewing Editor and Naama Barkai as the Senior Editor. One of the reviewers, Martin A Nowak, has agreed to share his name.

The reviewers have discussed the reviews with one another and the Reviewing Editor has drafted this decision to help you prepare a revised submission.

Essential revisions:

1) All the reviewers found the calculations extremely interesting and considered the results to be novel and important. However, there was a shared concern that the connection of these results with the biological phenomenon of the incubation period was not on firm ground. In particular, the assumption that 100% – or close to 100% – of the host cells are infected when symptomatic infection starts was not well-motivated, and the biological plausibility of this assumption was unclear.

2) We would like to encourage the authors to reconsider the interpretation of their findings. Ideally they should provide a broader a list of biological phenomena (not only incubation periods) that could be described by their model.

3) If they want to strengthen their interpretation of incubation periods, a mechanistic description how the model describes specific diseases is needed along with a discussion of possible caveats.

4) It would also be of interest to discuss the question of time-varying N (either growing N, or declining N as a result of the pathogen killing host cells). However, following consultation, it was felt that anything more than discussion of this question is beyond the scope of the current work.

*Reviewer #1:*

Strogatz et al. present a model for why the length of disease incubation periods tend to have a right-skewed probability distribution. The core idea is marvelously simple: even in the absence of complications such as host or pathogen heterogeneity, a simple stochastic model for within-host disease spread predicts a right-skewed distribution for the time between inoculation and the pathogen reaching a threshold value (corresponding to the threshold of immune activation that marks the end of the incubation period). The main text is unusually clearly written for a pure theory paper.

My primary concerns surround the relatively limited connection back to the biology. The authors consider multiple different possibilities for within-host network connectivity and provide some biological justification for the different topologies (e.g., structured tissue = 3d lattice, epithelium = 2d lattice, etc.); however:

1) The analysis of the Moran model focuses on the time to fixation of the invading allele (i.e. 100% infection of the modeled network); however, immune activation does not wait until 100% of the population is infected but rather activates at a much lower threshold (admittedly again in a stochastic fashion). One of the key findings of this work is that a right-skewed distribution arises due to the "coupon collector problem" where the last few uninfected nodes in the graph can take a long time to be infected, particularly for high-dimensional topologies. However, if the incubation period were to end when, say, any 1% of cells were infected rather than 100%, then the coupon collector problem seems less (or perhaps not at all) relevant.

2) Infection / reproduction in a spatial situation will often happen in parallel rather than serially. The argument that high-dimensional topology may lead to a slow down of the final infections appears predicated on the idea that the rate of infection is constant due to the serial nature of the Moran model – but the overall infection rate presumably increases as a function of the surface area of the infected volume (and analogously for other topologies). Does this really not matter for the distribution of time to fixation? This area is not my specialty, but the theory surrounding Fisher's wave of advance may be relevant.

3) Many pathogens cause infections in a bursting fashion (e.g., lytic viruses) for which it is not obvious if the Moran model assumption of one random birth / one random death at a time is relevant.

*Reviewer #2:*

Ottino-Loeffler et al. propose a simple and elegant solution, based on invasion dynamics in structured population, to the interesting observation that incubation periods for a variety of diseases are right-skewed. I found the idea to be very elegant, the text well-written, and overall the arguments to be compelling and easy to follow. I especially liked the interpretation for the dispersal coefficients naturally ranging between certain limits. However, I have a few concerns/questions that I would like to see addressed:

1) In order to apply the ideas of evolutionary graph theory the authors assume that the populations are finite (all graphs have only N nodes); however, this is not always true (certainly not for all the diseases that the authors mention in the abstract and introduction) and I would have liked to have seen from the authors at the very least an acknowledgement of this strong assumption and a discussion of how relaxing this assumption might affect the conclusions. It would be even better (and really interesting) if the authors could pick one example of dynamically-growing network (this should be doable at least for the complete graph) and see how that affects their predictions.

2) The Death-birth (DB) dynamic that the authors employ is actually different from the DB dynamic proposed by Ohtsuki et al. 2006 (unlike the BD dynamic which is the same). Ohtsuki et al. consider death to be random (all nodes have probability 1/N) and birth competition to be among the neighbors, proportional to their fitness. Of course, it's no problem proposing a new variant; I'm just curious whether the authors had a biological reason for choosing this variant of the DB update rule. Especially since they find in their Materials and methods (section on truncation) that the update rule actually matters a lot, a fact that has been observed in evolutionary graph theory more broadly. On that note, I thought this result was sufficiently important that it deserved at least a couple of sentences in the Discussion (rather than just being mentioned in the Materials and methods); I had the same reaction to all the results in that section ("Testing robustness to update rule, fitness, and truncation"), which I thought deserved some mention in the main Discussion.

3) My final point is more a question than a concern: the authors apply this method to in-host dynamics and incubation periods; however, it seems like it could apply to epidemiological questions as well (e.g. spread of flu in a population). Have the authors considered the parallels? Are there any data analogous to incubation periods that could be employed to show the applicability of this model to epidemiological questions as well?

*Reviewer #3:*

In Evolutionary Dynamics of Incubation Periods, Ottino-Loffler et al. investigate the distribution of times to fixation and to near fixation of mutants on an evolutionary graph as a model for incubation periods. Using this conceptual framework, they endeavor to explain features of the observed distribution of incubation times, including approximate log-normality with dispersion in the range of 1.1-1.5, without invoking heterogeneity in invader fitness, disease burden at which symptoms manifest, or initial dosage.

Ottino-Loffler et al. provide analytical calculations for the distribution of fixation times on complete, star, and cycle graphs in the infinite-fitness mutant limit, as well as numerical simulation results on a variety of other graphs. The distribution of times to fixation for a mutant in a structured population is an extremely important and interesting theoretical question. The authors provide novel and fascinating mathematical results.

I am very much in favor of publication of these findings. I do have some questions or concerns regarding the biological interpretation of the paper.

What is the relationship between the model, specifically time to fixation in a Moran process on a graph, and the biological phenomenon of an incubation period. Clarification about the biological motivation for this theoretical framework and for the different update rules could strengthen the paper. In particular, which of the several diseases mentioned in the paper does the model apply to, and why? Also, clarification regarding the choice of fixation time to quantify the length of the incubation period would strengthen the paper. Lowering the threshold to below 100 percent appears to remove the skew in some instances.

Second, might the distribution of clock times differ from the distribution of step times in the process.

Third, certain model choices produce symmetric, rather than skewed, distributions of times to a threshold. I am curious if a more complete investigation of this observation could shed light on the appearance of skewness in different evolutionary scenarios.

A final small, technical clarification would help to understand the derivation of skewness when r = 1 at the end of the paper. How is the number of steps which do not change the number of mutants being counted? Further, at the end of the fourth paragraph of the subsection “Asymptotic skew of conditioned random walk”, should An(1) be Mn(1)?

Again I wish to emphasize the great interest and novelty of this work. While some issues associated with the interpretation of the paper remain unclear, I think that the authors will be able to address them by pointing to biological scenarios, perhaps even outside of infection, in which their calculations provide valuable insight.

---

## [Author Response]

Essential revisions:1) All the reviewers found the calculations extremely interesting and considered the results to be novel and important. However, there was a shared concern that the connection of these results with the biological phenomenon of the incubation period was not on firm ground. In particular, the assumption that 100% – or close to 100% – of the host cells are infected when symptomatic infection starts was not well-motivated, and the biological plausibility of this assumption was unclear.

We agree, and have addressed this point in a new paragraph added to Results, called “Testing robustness to update rule and truncation.” The relevant sentences are: “For example, suppose we allow symptoms to occur when invaders take over only a fraction *f* of the whole network. This is a reasonable consideration as leukemic cells need not take over all the bone marrow before leukemia becomes evident, nor does typhoid need to overwhelm all the cells in the microbiome before causing fever; indeed it is likely far fewer in both cases.” Figure 6) show that the right-skewed distribution persists for Death-birth dynamics, even if only 10% takeover triggers the appearance of symptoms. (Actually, the right-skewed distribution can be proven to persist for any fixed, nonzero fraction *f*. We cite the relevant math literature in the text.)

On the other hand, Birth-death dynamics are sensitive to the choice of *f*. They produce a distribution that switches from a Gumbel when *f* = 100% to a normal distribution for any *f* less than 100%.

We explain why the two cases differ in the caption to Figure 6. “The difference in sensitivity between the two types of dynamics can be explained intuitively by when coupon collection occurs. […] In contrast, coupon collection occurs near the *end* of the invasion for Birth-death dynamics (when residents are scarce), and hence gets truncated when *f* < 1, giving rise to a normal instead of a right-skewed distribution.”

2) We would like to encourage the authors to reconsider the interpretation of their findings. Ideally they should provide a broader a list of biological phenomena (not only incubation periods) that could be described by their model.

We appreciate this helpful suggestion. In response, we have added the following paragraph at the end of the Discussion: “Aside from their possible application to incubation processes, our results also shed light on a broader theoretical question in evolutionary dynamics: when a mutant invades a structured population of residents, how does the distribution of mutant fixation times depend on the network structure of the population? […] We hope that our exact results for disparate topologies and dynamics will stimulate further investigations of these important questions in evolutionary biology.”

3) If they want to strengthen their interpretation of incubation periods, a mechanistic description how the model describes specific diseases is needed along with a discussion of possible caveats.

We have tried to connect the model more closely to specific diseases. In the section of the Results called “Mathematical model,” the relevant passage reads: “A network of *N* ≫ 1 nodes is used to represent an environment within a host where a pathogenic agent, such as a bacterium or cancer cell, is invading and reproducing. The network could represent several plausible biological scenarios, for example the intestinal microbiome, where harmful typhoid bacteria are competing against a benign resident population of gut flora in a mixing system (modeled as a complete graph); or it could represent mutated leukemic stem-cells vying for space against healthy hematopoietic stem cells within the well-organized three-dimensional bone marrow space (modeled as a 3D lattice); or a flat epithelial sheet with an early squamous cancer compromising and invading nearby healthy cells (modeled as a 2D lattice).”

Regarding possible caveats, we have clarified (in Results, “Mathematical model”) that the model probably does not apply to viruses:

“While Sartwell’s law has been applied to diseases as varied as measles and leukemia, the model we propose makes the most sense for asexually reproducing invaders, like cancer cells or bacteria. […] So, while the general phenomenon of network invasion seems to apply to viruses as well, this model is not well suited to describe their dynamics.”

We also draw a closer connection to cancer dynamics in the Discussion: “On the other hand, it is tempting to speculate that this regime of nearly neutral fitness may be more relevant to cancer development. […] This is also consistent with the suggestion of (Williams and Bjerknes, 1972); the shape of tumors in the model most closely resembled that of real tumors when the fitness of the invaders was only slightly above to neutral.”

4) It would also be of interest to discuss the question of time-varying N (either growing N, or declining N as a result of the pathogen killing host cells). However, following consultation, it was felt that anything more than discussion of this question is beyond the scope of the current work.

We are also interested in the effect of time-varying N, but also agree that a full investigation is outside of the scope of this paper. To give a preliminary sense of what might occur, we have added a new section in the Materials and methods, “Influence of non-static population,” as well as an additional figure (Figure 9). These show the results of an initial exploration of the effects of a non-static population on the complete graph. We specifically examine cases where the resident population has a chance at growing at every time step, shrinking at every time step, and randomly growing or shrinking at each time step.

Reviewer #1:

[…] My primary concerns surround the relatively limited connection back to the biology. The authors consider multiple different possibilities for within-host network connectivity and provide some biological justification for the different topologies (e.g., structured tissue = 3d lattice, epithelium = 2d lattice, etc.); however:1) The analysis of the Moran model focuses on the time to fixation of the invading allele (i.e. 100% infection of the modeled network); however, immune activation does not wait until 100% of the population is infected but rather activates at a much lower threshold (admittedly again in a stochastic fashion). One of the key findings of this work is that a right-skewed distribution arises due to the "coupon collector problem" where the last few uninfected nodes in the graph can take a long time to be infected, particularly for high-dimensional topologies. However, if the incubation period were to end when, say, any 1% of cells were infected rather than 100%, then the coupon collector problem seems less (or perhaps not at all) relevant.

Please see our response to Essential revisions comment 1.

2) Infection / reproduction in a spatial situation will often happen in parallel rather than serially. The argument that high-dimensional topology may lead to a slow down of the final infections appears predicated on the idea that the rate of infection is constant due to the serial nature of the Moran model – but the overall infection rate presumably increases as a function of the surface area of the infected volume (and analogously for other topologies). Does this really not matter for the distribution of time to fixation? This area is not my specialty, but the theory surrounding Fisher's wave of advance may be relevant.

In the models presented, the overall infection rate does indeed grow with the surface area of the infected population. This has been clarified in Materials and methods, “Birth-death, other solvable networks,” with the following sentences: “For a two-dimensional square lattice, it is more difficult to produce analytical results that are both rigorous and exact. But by making an approximation based on the geometry of the lattice, and using the fact that the population growth rate is proportional to its surface area (see the Appendix, "Normally distributed fixation times for 2D lattice"), we can make a non-rigorous analytical guess about the distribution of the fixation times *T*.”

This is also elaborated upon in the Appendix, "Normally distributed fixation times for 2D lattice," with: “Assuming this to be true, recall the basic dynamics of infinite-*r* Bd with *N* nodes and *m* invaders. First, we uniformly select one node out of the population of invaders, which is always a probability of 1/*m* per node. Then we replace one of the invader's neighbors, uniformly at random. However, only invaders on the surface of the cluster even have a chance at replacing a healthy node!

Given sufficient regularity of the boundary of the cluster, this means that the probability of an invader replacing a healthy node is proportional to

qm=1m⋅

3) Many pathogens cause infections in a bursting fashion (e.g., lytic viruses) for which it is not obvious if the Moran model assumption of one random birth / one random death at a time is relevant.

We agree, and we have now clarified (in Results, “Mathematical model”) that the model probably does not apply to viruses: “While Sartwell's law has been applied to diseases as varied as measles and leukemia, the model we propose makes the most sense for asexually reproducing invaders, like cancer cells or bacteria. […] So, while the general phenomenon of network invasion seems to apply to viruses as well, this model is not well suited to describe their dynamics.”

Reviewer #2:

Ottino-Loeffler et al. propose a simple and elegant solution, based on invasion dynamics in structured population, to the interesting observation that incubation periods for a variety of diseases are right-skewed. I found the idea to be very elegant, the text well-written, and overall the arguments to be compelling and easy to follow. I especially liked the interpretation for the dispersal coefficients naturally ranging between certain limits. However, I have a few concerns/questions that I would like to see addressed:1) In order to apply the ideas of evolutionary graph theory the authors assume that the populations are finite (all graphs have only N nodes); however, this is not always true (certainly not for all the diseases that the authors mention in the abstract and introduction) and I would have liked to have seen from the authors at the very least an acknowledgement of this strong assumption and a discussion of how relaxing this assumption might affect the conclusions. It would be even better (and really interesting) if the authors could pick one example of dynamically-growing network (this should be doable at least for the complete graph) and see how that affects their predictions.

Please see our response to Essential revisions comment 4.

2) The Death-birth (DB) dynamic that the authors employ is actually different from the DB dynamic proposed by Ohtsuki et al. 2006 (unlike the BD dynamic which is the same). Ohtsuki et al. consider death to be random (all nodes have probability 1/N) and birth competition to be among the neighbors, proportional to their fitness. Of course, it's no problem proposing a new variant; I'm just curious whether the authors had a biological reason for choosing this variant of the DB update rule. Especially since they find in their Materials and methods (section on truncation) that the update rule actually matters a lot, a fact that has been observed in evolutionary graph theory more broadly. On that note, I thought this result was sufficiently important that it deserved at least a couple of sentences in the Discussion (rather than just being mentioned in the Materials and methods); I had the same reaction to all the results in that section ("Testing robustness to update rule, fitness, and truncation"), which I thought deserved some mention in the main Discussion.

To clarify this, we have added a feature box ([Box box2], “Nomenclature for the Moran Model”), which includes an explanation and table describing and naming the most common Moran models. In particular, we would call the model described by the reviewer “dB,” as opposed to the Db model we used. The relevant sentences are: “To avoid confusion, we use standard abbreviations to distinguish the different models, as illustrated by the table below. […] For example, dB refers to the update rule where the first step uniformly selects a node from the entire population to die, and then one of its neighbors is selected, with probability proportional to fitness, to replace it.”

3) My final point is more a question than a concern: the authors apply this method to in-host dynamics and incubation periods; however, it seems like it could apply to epidemiological questions as well (e.g. spread of flu in a population). Have the authors considered the parallels? Are there any data analogous to incubation periods that could be employed to show the applicability of this model to epidemiological questions as well?

We find this possibility interesting, and have previously considered the parallels. However, a full investigation into the particulars of epidemic dynamics would be beyond the scope of the current paper. Nonetheless a connection can likely be made via evolutionary graph theory, which we mention in a new paragraph added to the Discussion: “Aside from their possible application to incubation processes, our results also shed light on a broader theoretical question in evolutionary dynamics: when a mutant invades a structured population of residents, how does the distribution of mutant fixation times depend on the network structure of the population? […] We hope that our exact results for disparate topologies and dynamics will stimulate further investigations of these important questions in evolutionary biology.”

Reviewer #3:

[...] I am very much in favor of publication of these findings. I do have some questions or concerns regarding the biological interpretation of the paper.What is the relationship between the model, specifically time to fixation in a Moran process on a graph, and the biological phenomenon of an incubation period. Clarification about the biological motivation for this theoretical framework and for the different update rules could strengthen the paper. In particular, which of the several diseases mentioned in the paper does the model apply to, and why?

Please see our response to Essential revisions comment 3.

Also, clarification regarding the choice of fixation time to quantify the length of the incubation period would strengthen the paper. Lowering the threshold to below 100 percent appears to remove the skew in some instances.

Please see our response to Essential revisions comment 1.

Second, might the distribution of clock times differ from the distribution of step times in the process.

To clarify this important point, we have added a new paragraph to the Results section, “Mathematical model.” The relevant paragraph reads: “Our notion of time in this model is linked directly to the biology of invasion of a reproducing asexual pathogen that divides and replaces other cells sequentially. […] Future iterations of this model could consider deriving an exact scaling between physical time and this biological event-based updating of time.”

Third, certain model choices produce symmetric, rather than skewed, distributions of times to a threshold. I am curious if a more complete investigation of this observation could shed light on the appearance of skewness in different evolutionary scenarios.

This is a fascinating question. A universal condition to ensure the appearance of symmetric distributions is beyond the scope of this paper, but the current conditions appear to involve infinite invader fitness combined with either a low-dimensional network or a truncation condition that avoids the Coupon Collector’s effect.

A final small, technical clarification would help to understand the derivation of skewness when r = 1 at the end of the paper. How is the number of steps which do not change the number of mutants being counted?

For the sake of generality, the specific calculation mentioned here calculates the skew that arises from the random walk alone, not including the waiting times. In the Appendix section “Asymptotic skew of conditioned random walk,” the following sentences have been added to clarify and justify this choice: “We will use the wait-omitted time *n* in this section as a first-order approximation of the true takeover time. […] Moreover, scaling and numerical arguments based on the results here can show that the bulk of the final distribution is defined by this random-walk process.”

Further, at the end of the fourth paragraph of the subsection “Asymptotic skew of conditioned random walk”, should An(1) be Mn(1)?

Thank you, this has been corrected.

Again I wish to emphasize the great interest and novelty of this work. While some issues associated with the interpretation of the paper remain unclear, I think that the authors will be able to address them by pointing to biological scenarios, perhaps even outside of infection, in which their calculations provide valuable insight.

We have added a paragraph about the model’s wider relevance to evolutionary dynamics: “Aside from their possible application to incubation processes, our results also shed light on a broader theoretical question in evolutionary dynamics: when a mutant invades a structured population of residents, how does the distribution of mutant fixation times depend on the network structure of the population? […] We hope that our exact results for disparate topologies and dynamics will stimulate further investigations of these important questions in evolutionary biology.”